# Learning in Congestion Games with Bandit Feedback

**Qiwen Cui**[1]*
qwcui@cs.washington.edu

**Zhihan Xiong**[1]*
zhihanx@cs.washington.edu

**Maryam Fazel**[2]
mfazel@uw.edu

**Simon S. Du**[1]
ssdu@cs.washington.edu

[1] Paul G. Allen School of Computer Science & Engineering, University of Washington
[2] Department of Electrical & Computer Engineering, University of Washington

## Abstract

In this paper, we investigate Nash-regret minimization in congestion games, a class of games with benign theoretical structure and broad real-world applications. We first propose a centralized algorithm based on the optimism in the face of uncertainty principle for congestion games with (semi-)bandit feedback, and obtain finite-sample guarantees. Then we propose a decentralized algorithm via a novel combination of the Frank-Wolfe method and G-optimal design. By exploiting the structure of the congestion game, we show the sample complexity of both algorithms depends only polynomially on the number of players and the number of facilities, but not the size of the action set, which can be exponentially large in terms of the number of facilities. We further define a new problem class, Markov congestion games, which allows us to model the non-stationarity in congestion games. We propose a centralized algorithm for Markov congestion games, whose sample complexity again has only polynomial dependence on all relevant problem parameters, but not the size of the action set.

## 1 Introduction

Nash equilibrium (NE) is a widely adopted concept in game theory community, used to describe the behavior of multi-agent systems with selfish players [Roughgarden, 2010]. At the Nash equilibrium, no player has the incentive to change its own strategy unilaterally, which implies it is a steady state of the game dynamics. For a general-sum game, computing the Nash equilibrium is PPAD-hard [Daskalakis, 2013] and the query complexity is exponential in the number of players [Rubinstein, 2016]. To help address these issues, a natural approach is to consider games with special structures. In this paper, we focus on congestion games.

Congestion games are general-sum games with *facilities* (resources) shared among noncooperative players [Rosenthal, 1973]. During the game, each player will decide what combination of facilities to utilize, and popular facilities will become congested, which results in a possibly higher cost on each user. One example of congestion game is the routing game [Fotakis et al., 2002], where each player needs to travel from a given starting point to a destination point through some shared routes. These routes are represented as a traffic graph and the facilities are the edges. Each player will decide her path to go, and the more players use the same edge, the longer the edge travel time will be. Congestion games also have wide applications in electrical grids [Ibars et al., 2010], internet routing

---

*Equal contribution.

36th Conference on Neural Information Processing Systems (NeurIPS 2022).

[Al-Kashoash et al., 2017] and rate allocation [Johari and Tsitsiklis, 2004]. In many real-world scenarios, players can only have (semi-)bandit feedback, i.e., players know only the payoff of the facilities they choose. This kind of learning under uncertainty has been widely studied in bandits and in reinforcement learning for the single-agent setting, while theoretical understanding for the multi-agent case is still largely missing.

There are two types of algorithms in multi-agent systems, namely centralized algorithms and decentralized algorithms. For centralized algorithms, there exists a central authority that can control and receive feedback from all players in the game. As we have global coordination, centralized algorithms usually have favorable performance. On the other hand, such a central authority may not always be available in practice, and thus people turn to decentralized algorithms, i.e., each player makes decisions individually and can only observe her own feedback. However, decentralized algorithms are vulnerable to *nonstationarity* because each player is making decisions in a nonstationary environment as others' strategies are changing [Zhang et al., 2021a]. In this paper, we will study both centralized and decentralized algorithms in congestion games with bandit feedback, and we will provide motivating scenarios for both algorithms in Section 1.2.

The main challenge in designing algorithms for $m$-player congestion games with bandit feedback is the curse of exponential action set, i.e., the number of actions can be exponential in the number of facilities $F$ because every subset of facilities can be an action. As a result, an efficient algorithm should have sample complexity polynomial in $m$ and $F$ and has no dependence on the size of the action space. One closely related type of general-sum game is the potential game, in which each individual's payoff changes, resulting from strategy modification, can be quantified by a common potential function. It is well-known that all congestion games are potential games, and each potential game has an equivalent congestion game formulation [Monderer and Shapley, 1996]. However, existing algorithms designed for potential games all have sample complexity scaling at least linearly in the number of actions [Leonardos et al., 2021, Ding et al., 2022], which is inefficient for congestion games. This motivates the following question:

*Can we design provably sample-efficient centralized and decentralized learning algorithms for congestion games with bandit feedback?*

We provide an affirmative answer to this question. To be precise, we use Nash-regret minimization (formally defined in Section 3) as our objective for learning in congestion games. This regret-like objective commonly appears in the literature of online learning and reinforcement learning [Orabona, 2019, Ding et al., 2022, Liu et al., 2021], which focuses on finite-time analysis and accumulative rewards throughout the learning process instead of the asymptotic behavior. In general, a sublinear Nash regret implies a best-iterate convergence, meaning that the algorithm has reached the approximate Nash equilibrium at least once, while the converse does not hold.

We highlight our contributions below and compare our results with previous algorithms in Table 1. Our algorithms are shaded and we prove sublinear Nash regrets for all of them. In Table 1, sample complexity refers to the number of samples required to reach best-iterate convergence to an $\epsilon$-approximate Nash equilibrium and the results are obtained by standard online-to-batch conversion as in Section 3.1 of [Jin et al., 2018].

## 1.1 Main Novelties and Contributions

**1. Centralized algorithm for congestion game.** We adapt the principle of optimism in the face of uncertainty in stochastic bandits to ensure sufficient exploration in congestion games. We begin with congestion games with semi-bandit feedback, in which each player can observe the reward of every facility in the action. Instead of estimating the action reward as in stochastic multi-armed bandits, we estimate the facility rewards directly, which *removes the dependence on the size of action space*. Furthermore, we consider congestion games with bandit feedback, in which each player can only observe the overall reward. In this setting, we borrow ideas from linear bandits to estimate the reward function and analyze the algorithm. The algorithm is provably sample efficient in both cases.

**2. Decentralized algorithm for congestion game.** Our decentralized algorithm is a Frank-Wolfe method with exploration, in which each player only observes her own actions and rewards. To efficiently explore in the congestion game, we utilize G-optimal design allocation for bandit feedback and a specific distribution for semi-bandit feedback. As a result, the sample complexity does not depend on the number of actions. In addition, the $L_1$ smoothness parameter of the potential function

Table 1: Comparison of algorithms for congestion games in terms of sample complexity and Nash regret, where "IPPG" stands for "independent projected policy gradient", "IPGA" stands for "independent policy gradient ascent", "I" represents the setting of semi-bandit feedback and "II" represents the setting of bandit feedback. Bandit feedback is assumed for algorithms from previous work. Here, $A_i$ is the size of player $i$'s action space, $m$ is the number of players, $A_{\max} = \max_{i \in [m]} A_i$, $F$ is the number of facilities and $T$ is the number of samples collected. Our algorithms are shaded.

| Algorithms | Sample complexity | Nash regret | Decentralized |
|---|---|---|---|
| Nash-VI [Liu et al., 2021] | $(\prod_{i=1}^{m} A_i)F/\epsilon^2$ | $\sqrt{(\prod_{i=1}^{m} A_i)FT}$ | No |
| V-learning [Jin et al., 2021a] | $A_{\max}F/\epsilon^2$ (CCE) | NA | Yes |
| IPPG [Leonardos et al., 2021] | $A_{\max}mF/\epsilon^6$ | NA | Yes |
| IPGA [Ding et al., 2022] | $A_{\max}^2 m^3 F^5/\epsilon^5$ | $mF^{4/3}\sqrt{A_{\max}}T^{4/5}$ | Yes |
| Nash-UCB I | $mF^2/\epsilon^2$ | $F\sqrt{mT}$ | No |
| Nash-UCB II | $m^2F^3/\epsilon^2$ | $mF^{3/2}\sqrt{T}$ | No |
| Frank-Wolfe with Exploration I | $m^{12}F^9/\epsilon^6$ | $m^2F^{3/2}T^{5/6}$ | Yes |
| Frank-Wolfe with Exploration II | $m^{12}F^{12}/\epsilon^6$ | $m^2F^2T^{5/6}$ | Yes |

does not depend on the number of actions, which is exploited by the Frank-Wolfe method. With the help of these two specific algorithmic designs for congestion games, we give the first decentralized algorithm for both semi-bandit feedback and bandit feedback that has no dependence on the size of the action space in congestion games.

**3. Centralized algorithm for independent Markov congestion game.** We extend the formulation of congestion game into a Markov setting and propose the independent Markov congestion game (IMCG), in which each facility has its own internal state and state transition happens independently among all the facilities. In Section 1.2, we give some examples that fit in this model. By utilizing techniques from factored MDPs, we extend our centralized algorithms for congestion games to efficiently solve IMCGs, with both semi-bandit and bandit feedback.

## 1.2 Motivating Examples

We provide an exmple here to motivate our proposed models. See Section 3 for the formal definition of (semi-)bandit feedback and (Markov) congestion games and Appendix A for additional examples.

*Example* 1 (**Routing Games**). For a routing game, there are multiple players in a traffic graph travelling from starting points to destination points, and the facilities are the edges (roads). The cost of each edge is the waiting time, which depends on the number of players using that edge.

• **Centralized algorithm for routing games:** Imagine each player is using Google Maps to navigate. Then Google Maps can serve as a center that knows the starting points and the destination points, as well as the real-time feedback of the waiting time on each edge of all the players. Google Maps itself also has the incentive to assign paths according to the Nash equilibrium strategy as then each player will find out that deviating from the navigation has no benefit and thus sticks to the app.

• **Decentralized algorithm for routing games:** Consider the case where players are still using Google Maps but due to privacy concerns or limited bandwidth, they only use the offline version, which has access only to the information of each single user. Then Google Maps needs to use decentralized algorithms so that it can still assign Nash equilibrium strategy to each user after repeated plays.

• **Markov routing games:** For Markov routing games, the time cost on each edge will change between different timesteps, which is a more accurate model of the real-world. For instance, some roads are prone to car accidents, which will result in an increasing cost on the next timestep, and the chance of accidents also depends on the number of players using that edge currently. This is modeled by the Markovian facility state transition in independent Markov congestion games.

## 2 Related Work

**Potential Games.** Potential games are general-sum games that admit a common potential function to quantify the changes in individual's payoff [Monderer and Shapley, 1996]. Algorithmic game

theory community has studied how different dynamics converge to the Nash equilibrium, e.g., best response dynamics [Durand, 2018, Swenson et al., 2018] and no-regret dynamics [Heliou et al., 2017, Cheung and Piliouras, 2020], while usually they provide only asymptotic convergence, with either full information setting or bandit feedback setting. Recently, reinforcement learning community studied Markov potential games with bandit feedback, which can be applied to standard potential games. See the Markov Games part below for more details.

**Congestion Games.** Congestion games are developed in the seminal work [Rosenthal, 1973], and later Monderer and Shapley [1996] builds a close connection between congestion games and potential games. Congestion games are divided into atomic and non-atomic congestion games depending on whether each player is separable. Many papers consider non-atomic congestion games with non-decreasing cost function, which implies a convex potential function [Roughgarden and Tardos, 2004]. We consider the more difficult atomic congestion game where the potential function can be non-convex. For online non-atomic case, [Krichene et al., 2015] considers partial information setting while they provide convergence in the sense of Cesaro means. [Kleinberg et al., 2009, Krichene et al., 2014] show that some no-regret online learning algorithms asymptotically converges to Nash equilibrium. [Chen and Lu, 2015, 2016] are two closely related works that consider bandit feedback in atomic congestion games and provide non-asymptotic convergence. However, they still assume a convex potential function and the sample complexity has exponential dependence on the number of facilities, which is far from ideal.

**Markov Games.** Markov games are widely studied since the seminal work [Shapley, 1953]. Recently, the topic has received much attention due to advances in reinforcement learning theory. Liu et al. [2021] provides a centralized algorithm for learning the Nash equilibrium in general-sum Markov games, and [Jin et al., 2021a, Song et al., 2021] provide decentralized algorithms for learning the (coarse) correlated equilibrium. One closely related line of research is on Markov potential games [Leonardos et al., 2021, Zhang et al., 2021b, Fox et al., 2021, Cen et al., 2022, Ding et al., 2022]. However, applying their algorithms to congestion games leads to explicit dependence on the number of actions, which would be exponentially worse than our algorithms. See Table 1 for comparisons. Our independent Markov congestion game is motivated by the state-based potential games studied in Marden [2012] and Macua et al. [2018], and its transition kernel is closely related to the factored MDPs, for which single agent algorithms are studied in [Osband and Van Roy, 2014, Chen et al., 2020, Xu and Tewari, 2020, Tian et al., 2020, Rosenberg and Mansour, 2021].

**Learning in Games.** Different from our paper, learning in games in traditional literature of game theory mainly considers players' asymptotic behavior [Leslie and Collins, 2005, Cominetti et al., 2010, Coucheney et al., 2015]. In early literature, Leslie [2004] investigates actor-critic learning and $Q$-learning algorithms in games with bandit feedback and their connection to best-response dynamics. Leslie and Collins [2005] proposes individual $Q$-learning algorithm and shows that it converges to the NE almost surely in two-player zero-sum game and Leslie and Collins [2006] studies learning the NE from the perspective of a fictitious play-like process. Later, Cominetti et al. [2010] considers payoff-based learning rules and shows convergence to NE in traffic games, while another payoff-based learning model for continuous games is developed in Bervoets et al. [2020]. Coucheney et al. [2015] derives a new penalty-regulated dynamics and proposes a corresponding learning algorithms that converges to NE in potential games with bandit feedback. Bravo et al. [2018] proposes that in monotone games with bandit feedback, as long as all players are using some no-regret learning algorithm, the dynamics will converge to the NE, and an improved analysis of the same derivative-free algorithm is given in Drusvyatskiy et al. [2022]. In contrast, our learning objective focuses on finite-time cumulative rewards, which is more widely used in current multi-agent reinforcement learning literature [Ding et al., 2022, Liu et al., 2021].

## 3 Preliminaries

**General-sum Matrix Games.** We consider the model of general-sum matrix games, defined by the tuple $\mathcal{G} = (\{\mathcal{A}_i\}_{i=1}^m, R)$, where $m$ is the number of players, $\mathcal{A}_i$ is the action space of player $i$ and $R(\cdot|\boldsymbol{a})$ is the reward distribution on $[0, r_{\max}]^m$ with mean $\boldsymbol{r}(\boldsymbol{a})$. Let $\mathcal{A} = \mathcal{A}_1 \times \cdots \times \mathcal{A}_m$ be the whole action space and denote an element as $\boldsymbol{a} = (a_1, \ldots, a_m) \in \mathcal{A}$. After all players take actions $\boldsymbol{a} \in \mathcal{A}$, a reward vector is sampled $\boldsymbol{r} \sim R(\cdot|\boldsymbol{a})$ and player $i$ will receive reward $r_i \in [0, r_{\max}]$ with mean $r_i(\boldsymbol{a})$. Each player's objective is to maximize her own reward.

A general policy $\pi$ is defined as a vector in $\Delta(\mathcal{A})$, the probability simplex over the action space $\mathcal{A}$. A product policy $\pi = (\pi_1, \ldots, \pi_m)$ is defined as a tuple in $\Delta(\mathcal{A}_1) \times \cdots \times \Delta(\mathcal{A}_m)$, in which $\boldsymbol{a} = (a_1, \ldots, a_m) \sim \pi$ represents $a_i \overset{\text{i.i.d.}}{\sim} \pi_i$. The value of policy $\pi$ for player $i$ is $V_i^\pi = \mathbb{E}_{\boldsymbol{a} \sim \pi}[r_i(\boldsymbol{a})]$.

**Nash Equilibrium and Nash Regret.** Given a general policy $\pi$, let $\pi_{-i}$ be the marginal joint policy of players $1, \ldots, i-1, i+1, \ldots, m$. Then, the best response of player $i$ under policy $\pi$ is $\pi_i^\dagger = \arg\max_{\mu \in \Delta(\mathcal{A}_i)} V_i^{\mu, \pi_{-i}}$ and the corresponding value is $V_i^{\dagger, \pi_{-i}} := V_i^{\pi_i^\dagger, \pi_{-i}}$. Our goal is to find the approximate Nash equilibrium of the matrix game, which is defined below.

**Definition 1.** A product policy $\pi$ is an $\epsilon$-*approximate Nash equilibrium* if $\max_i(V_i^{\dagger, \pi_{-i}} - V_i^\pi) \leq \epsilon$.

An $\epsilon$-approximate Nash equilibrium can be obtained by achieving a sublinear Nash regret, which is defined below. See Section 3 in Ding et al. [2022] for a more detailed discussion.

**Definition 2.** With $\pi^k$ being the policy at $k$-th episode, the *Nash regret* after $K$ episodes is define as

$$\text{Nash-Regret}(K) = \sum_{k=1}^{K} \max_{i \in [m]} \left( V_i^{\dagger, \pi_{-i}^k} - V_i^{\pi^k} \right).$$

*Remark* 1. Here, if we replace $\max_{i \in [m]}$ by $\sum_{i=1}^{m}$ in the definition of Nash regret, the single-step Nash regret at episode $k$ will become the Nikaido-Isoda (NI) function evaluated at $\pi^k$, which is a popular objective for equilibrium computation [Nikaidô and Isoda, 1955, Raghunathan et al., 2019]. Replacing $\max_{i \in [m]}$ by $\sum_{i=1}^{m}$ will multiply our regret bounds by a factor of $m$, while our conclusion will not be affected.

**Potential Games.** A potential game is a general-sum game such that there exists a potential function $\Phi : \Delta(\mathcal{A}) \to [0, \Phi_{\max}]$ such that for any player $i \in [m]$ and policies $\pi_i, \pi_i', \pi_{-i}$, it satisfies

$$\Phi(\pi_i, \pi_{-i}) - \Phi(\pi_i', \pi_{-i}) = V_i^{\pi_i, \pi_{-i}} - V_i^{\pi_i', \pi_{-i}}.$$

We can immediately see that a policy that maximizes the potential function is a Nash equilibrium.

**Congestion Games.** A congestion game is defined by $\mathcal{G} = (\mathcal{F}, \{\mathcal{A}_i\}_{i=1}^m, \{R^f\}_{f \in \mathcal{F}})$, where $\mathcal{F} = [F]$ is called the facility set and $R^f(\cdot|n) \in [0,1]$ is the reward distribution for facility $f$ with mean $r^f(n)$, where $n \in [m]$. Each action $a_i \in \mathcal{A}_i$ is a subset of $\mathcal{F}$ (i.e., $a_i \subseteq \mathcal{F}$). Suppose the joint action chosen by all the players is $\boldsymbol{a} \in \mathcal{A}$, then a random reward is sampled $r^f \sim R^f(\cdot|n^f(\boldsymbol{a}))$ for each facility $f$, where $n^f(\boldsymbol{a}) = \sum_{i=1}^m \mathbb{1}\{f \in a_i\}$ is the number of players using facility $f$. The reward collected by player $i$ is $r_i = \sum_{f \in a_i} r^f$ with mean $r_i(\boldsymbol{a}) = \sum_{f \in a_i} r^f(n^f(\boldsymbol{a})) \in [0, F]$.

**Connection to Potential Games [Monderer and Shapley, 1996].** As a special class of potential game, all congestion games have the potential function: $\Phi(\boldsymbol{a}) = \sum_{f \in \mathcal{F}} \sum_{i=1}^{n^f(\boldsymbol{a})} r^f(i)$. To see this, we can easily verify that $\Phi(a_i, a_{-i}) - \Phi(a_i', a_{-i}) = r_i(a_i, a_{-i}) - r_i(a_i', a_{-i})$ holds. Then, by defining $\Phi(\pi) = \mathbb{E}_{\boldsymbol{a} \sim \pi}[\Phi(\boldsymbol{a})]$, we can have $\Phi(\pi_i, \pi_{-i}) - \Phi(\pi_i', \pi_{-i}) = V_i^{\pi_i, \pi_{-i}} - V_i^{\pi_i', \pi_{-i}}$.

**Types of feedback.** There are in general two types of reward feedback for the congestion games, semi-bandit feedback and bandit feedback, both of which are reasonable under different scenarios. In semi-bandit feedback, after taking the action, player $i$ will receive reward information $r^f$ for each $f \in a_i$; in bandit feedback, after taking the action, player $i$ will only receive the reward $r_i = \sum_{f \in a_i} r^f$ with no knowledge about each $r^f$. In this paper, we will address both of them, with more focus on the bandit feedback, which can be directly generalized to semi-bandit feedback.

# 4 Centralized Algorithms for Congestion Games

In this section, we introduce two centralized algorithms for congestion games – one for the semi-bandit feedback and one for the bandit feedback. We will see that both of them can achieve sublinear Nash regret with polynomial dependence on both $m$ and $F$.

## 4.1 Algorithm for Semi-bandit Feedback

Summarized in Algorithm 1, Nash upper confidence bound (Nash-UCB) for congestion games is developed based on optimism in the face of uncertainty. In particular, the algorithm estimates the

reward matrices optimistically in line 4, computes its Nash equilibrium policy in line 5 and then follows this policy.

For convenience, we define the empirical counter $N^{k,f}(n) = \sum_{k'=1}^{k} \mathbb{1}\left\{n^f(\boldsymbol{a}^{k'}) = n\right\}$ and $\tilde{\iota} = 2\log(4(m+1)K/\delta)$. Then, the reward estimator for $f$ and the bonus term are defined as

$$\hat{r}^{k,f}(n) = \frac{\sum_{k'=1}^{k} r^{k',f} \mathbb{1}\left\{n^f(\boldsymbol{a}^{k'}) = n\right\}}{N^{k,f}(n) \vee 1}, \quad b_i^{k,\mathrm{r}}(\boldsymbol{a}) = \sum_{f \in a_i} \sqrt{\frac{\tilde{\iota}}{N^{k,f}(n^f(\boldsymbol{a})) \vee 1}}, \quad (1)$$

where $r^{k,f} \in [0,1]$ is the random reward realization of $r^f(n^f(\boldsymbol{a}^k))$. Naturally, the reward estimator for player $i$ is $\hat{r}_i^k(\boldsymbol{a}) = \sum_{f \in a_i} \hat{r}^{k,f}(n^f(\boldsymbol{a}))$.

---

**Algorithm 1** Nash-UCB for Congestion Games

1: **Input:** $\epsilon$, accuracy parameter for Nash equilibrium computation
2: **for** episode $k = 1, \ldots, K$ **do**
3:     **for** player $i = 1, \ldots, m$ **do**
4:         $\overline{Q}_i^k(\boldsymbol{a}) \leftarrow \hat{r}_i^k(\boldsymbol{a}) + b_i^{k,\mathrm{r}}(\boldsymbol{a})$ for all $\boldsymbol{a} \in \mathcal{A}$
5:     $\pi^k \leftarrow \epsilon\text{-NASH}(\overline{Q}_1^k(\cdot), \cdots, \overline{Q}_m^k(\cdot))$ (Algorithm 2)
6:     Take action $\boldsymbol{a}^k \sim \pi^k$ and observe reward $r^{k,f}$
7:     Update reward estimators $\hat{r}_i^k$ and bonus term $b_i^{k,\mathrm{r}}$

---

Algorithm 1 is motivated by the Nash-VI algorithm in [Liu et al., 2021] plus a deliberate utilization of the special reward structure in the congestion games. Moreover, notice that a matrix game with reward functions $\overline{Q}_1^k(\cdot), \ldots, \overline{Q}_m^k(\cdot)$ forms a potential game (see Lemma 1). As a result, in line 5, we can *efficiently compute* the $\epsilon$-approximate Nash equilibrium $\pi^k$ for that matrix game by utilizing Algorithm 2, (see Lemma 2). It is a simple greedy algorithm such that in each round, it modifies one player's policy whose modification can increase the potential function most. In addition, Algorithm 2 always outputs a deterministic product policy.

---

**Algorithm 2** $\epsilon$-approximate Nash Equilibrium for Potential Games

1: **Input:** $\epsilon$, accuracy parameter; full information potential game $(\{\mathcal{A}_i\}_{i=1}^m, \{r_i\}_{i=1}^m)$ such that $r_i \in [0, r_{\max}]$ for all $i \in [m]$
2: **Initialize:** $\pi^1 = \boldsymbol{a}^1$, arbitrary deterministic product policy
3: **for** round $k = 1, \ldots, \left\lceil \frac{m r_{\max}}{\epsilon} \right\rceil$ **do**
4:     **for** player $i = 1, \ldots, m$ **do**
5:         $\Delta_i = \max_{a_i \in \mathcal{A}_i} r_i(a_i, \pi_{-i}^k) - r_i(\pi^k)$
6:         $a_i^{k+1} = \arg\max_{a \in \mathcal{A}_i} r_i(a_i, \pi_{-i}^k) - r_i(\pi^k)$
7:     **if** $\max_{i \in [m]} \Delta_i \leq \epsilon$ **then**
8:         **return** $\pi^k$
9:     $j = \arg\max_{i \in [m]} \Delta_i$
10:     $\pi^{k+1}(j) = a_j^{k+1}$, $\pi^{k+1}(i) = \pi^k(i)$, for all $i \neq j$

---

## 4.2 Algorithm for Bandit Feedback

When the players can only receive bandit feedback, estimating $\hat{r}^{k,f}$ directly for each $f \in \mathcal{F}$ is no longer feasible. However, notice that the reward function $r_i(\boldsymbol{a}) = \sum_{f \in a_i} r^f(n^f(\boldsymbol{a}))$ can be seen as an inner product between vectors characterized by action $\boldsymbol{a}$ and reward function $r^f(\cdot)$. Therefore, under bandit feedback, we can treat it as a linear bandit and use ridge regression to build the reward estimator $\tilde{r}_i^k$ and corresponding bonus term $\tilde{b}^{k,\mathrm{r}}$, whose index $i$ is dropped since it is the same for all players. The new algorithm will use these two terms to replace $\hat{r}_i^k$ and $b_i^{k,\mathrm{r}}$ in line 4 of Algorithm 1.

In particular, define $\theta \in [0,1]^{\tilde{d}}$ with $\tilde{d} = mF$ to be the vector such that $r^f(n) = \theta_{n+m(f-1)}$. Meanwhile, for player $i \in [m]$, define $A_i : \mathcal{A} \mapsto \{0,1\}^{\tilde{d}}$ to be the vector-valued function such that

$$[A_i(\boldsymbol{a})]_j = \mathbb{1}\left\{j = n + m(f-1), f \in a_i, n = n^f(\boldsymbol{a})\right\}.$$

In other words, $A_i(\boldsymbol{a})$ is a 0-1 vector with element 1 only at indices corresponding to those in $\theta$ that represents $r^f(n)$ for $f \in a_i$ and $n = n^f(\boldsymbol{a})$. Now, with these definitions, the reward function can be written as $r_i(\boldsymbol{a}) = \langle A_i(\boldsymbol{a}), \theta \rangle$. Then, we build the reward estimator and the bonus term through ridge regression and corresponding confidence bound, which are defined as the following:

$$\tilde{r}_i^k(\boldsymbol{a}) = \left\langle A_i(\boldsymbol{a}), \widehat{\theta}^k \right\rangle, \quad \tilde{b}^{k,\mathrm{r}}(\boldsymbol{a}) = \max_{i \in [m]} \|A_i(\boldsymbol{a})\|_{(V^k)^{-1}} \sqrt{\tilde{\beta}_k}, \tag{2}$$

where $\widehat{\theta}^k = \left(V^k\right)^{-1} \sum_{k'=1}^{k-1} \sum_{i=1}^m A_i(\boldsymbol{a}^{k'}) r_i^{k'}$, $V^k = I + \sum_{k'=1}^{k-1} \sum_{i=1}^m A_i(\boldsymbol{a}^{k'}) A_i(\boldsymbol{a}^{k'})^\top$ and $\sqrt{\tilde{\beta}_k} = \sqrt{\tilde{d}} + \sqrt{F\tilde{d}\log\left(1 + \frac{mkF}{\tilde{d}}\right) + F\tilde{\iota}}$. Note that we cannot bound the sum of this bonus terms by directly applying the elliptical potential lemma. We instead prove its variant in Lemma 4.

### 4.3 Regret Analysis

The Nash regret bounds for the two versions of Algorithm 1 are formally presented in Theorem 1. The proof details are deferred to Appendix C.

**Theorem 1.** *Let $\epsilon = 1/K$. For congestion games with semi-bandit feedback, by running Algorithm 1 with reward estimator and bonus term in (1), with probability at least $1 - \delta$, we can achieve that*

$$\textit{Nash-Regret}(K) \leq \widetilde{\mathcal{O}}\left(F\sqrt{mK}\right).$$

*Furthermore, if we only have bandit feedback, then by running Algorithm 1 with reward estimator and bonus term in (2), with probability at least $1 - \delta$, we can achieve that*

$$\textit{Nash-Regret}(K) \leq \widetilde{\mathcal{O}}\left(mF^{3/2}\sqrt{K}\right).$$

*Remark* 2. Since each action is a subset of $\mathcal{F}$, the size of each player's action space can be $2^F$. As a result, directly applying Nash-VI in [Liu et al., 2021] leads to a regret bound exponential in $F$.

*Remark* 3. Note that we assume $r^f \in [0,1]$, which implies $r_i \in [0, F]$ for each player $i \in [m]$.

## 5 Decentralized Algorithms for Congestion Games

In this section, we present a decentralized algorithm for congestion games. Due to limited space, we only introduce the version of bandit feedback as in Section 4.2. The algorithmic details for the semi-bandit feedback setting are deferred into Appendix D.3. We will show that under both settings, even though each player can only observe her own actions and rewards, our decentralized algorithm still enjoys sublinear Nash regret with polynomial dependence on $m$ and $F$.

We first define the vector-valued function $\phi_i : \mathcal{A}_i \mapsto \{0,1\}^{F_i}$ to be the feature map of player $i$ such that $[\phi_i(a_i)]_f = \mathbb{1}\{f \in a_i\}$ for $a_i \in \mathcal{A}_i$ and $f \in \bigcup_{a_i \in \mathcal{A}_i} a_i$. Here, $F_i$ is the size of $\bigcup_{a_i \in \mathcal{A}_i} a_i \subseteq \mathcal{F}$ and we can immediately see that $F_i \leq F$ for any $i \in [m]$.

The core idea of our algorithm is that the Nash equilibrium can be found by reaching the stationary points of the potential function since all congestion games are potential games. Here, the UCB-like algorithms used in the centralized setting are not applicable because their policy computation requires value functions for all players (e.g., line 5 of Algorithm 1), which are not available in the decentralized setting. Summarized in Algorithm 3, the decentralized algorithm is developed based on the Frank-Wolfe method and has the following three major components.

**Gradient Estimator.** In line 7, the algorithm builds the estimator $\widehat{\nabla}_i^k \Phi$ defined in (4) by using the $\tau$ reward samples collected from line 5. Here, $\widehat{\nabla}_i^k \Phi$ estimates the gradient of potential function $\Phi$ with respect to the policy $\pi_i^k$. Recall that for a congestion game, we have $\Phi(\boldsymbol{a}) = \sum_{f \in \mathcal{F}} \sum_{i=1}^{n^f(\boldsymbol{a})} r^f(i)$

---

**Algorithm 3** Frank-Wolfe with Exploration for Congestion Game

---

1: **Input:** $\gamma, \nu$, mixture weights; $\pi_i^1$, initial policy.
2: **Initialize:** $\rho_i$, the G-optimal design for player $i$, defined in (5).
3: **for** episode $k = 1, \cdots, K$ **do**
4:     **for** round $t = 1, \cdots, \tau$ **do**
5:         Each player takes action $a_i^{k,t} \sim \pi_i^k$, observes reward $r_i^{k,t}$.
6:     **for** player $i = 1, \cdots, m$ **do**
7:         Compute $\widehat{\nabla}_i^k \Phi(a_i)$ by the formula in (4) for all $a_i \in \mathcal{A}_i$
8:         Compute $\widetilde{\pi}_i^{k+1} \leftarrow \text{argmax}_{\pi_i \in \Delta(\mathcal{A}_i)} \left\langle \pi_i, \widehat{\nabla}_i^k \Phi \right\rangle$
9:         Update $\pi_i^{k+1} \leftarrow (1-\gamma)(\nu \widetilde{\pi}_i^{k+1} + (1-\nu)\pi_i^k) + \gamma \rho_i$

---

and $\Phi(\pi) = \mathbb{E}_{\boldsymbol{a} \sim \pi}[\Phi(\boldsymbol{a})]$. Then we can define $\nabla_i \Phi := \nabla_{\pi_i} \Phi$ as a vector of dimension $|\mathcal{A}_i|$. For the component indexed by some $a_i \in \mathcal{A}_i$, we can see that $\Phi(\pi) = \pi_i(a_i) \mathbb{E}_{a_{-i} \sim \pi_{-i}}[r_i(a_i, a_{-i})] + \text{const}$, where const does not depend on $\pi_i(a_i)$. Therefore, we have

$$\nabla_i \Phi(a_i) = \mathbb{E}_{a_{-i} \sim \pi_{-i}}[r_i(a_i, a_{-i})] = \mathbb{E}_{a_{-i} \sim \pi_{-i}}\left[\sum_{f \in a_i} r^f(n^f(a_i, a_{-i}))\right] = \langle \phi_i(a_i), \theta_i(\pi) \rangle, \quad (3)$$

where $[\theta_i(\pi)]_f = \mathbb{E}_{a_{-i} \sim \pi_{-i}}\left[r^f(n^f(a_{-i}) + 1)\right]$. Meanwhile, the mean of the $t$-th reward that player $i$ received at episode $k$ satisfies

$$\mathbb{E}\left[r_i^{k,t} \mid \boldsymbol{a}^{k,t}\right] = r_i(\boldsymbol{a}^{k,t}) = \sum_{f \in a_i^{k,t}} r^f(n^f(\boldsymbol{a}^{k,t})) = \left\langle \phi_i(a_i^{k,t}), \theta_i^{k,t}(a_{-i}^{k,t}) \right\rangle,$$

where $[\theta_i^{k,t}(a_{-i}^{k,t})]_f = r^f(n^f(a_{-i}^{k,t}) + 1)$ and its mean is $[\theta_i(\pi^k)]_f$. Therefore, we can use linear regression to estimate $\theta_i(\pi^k)$. In particular, we have $\widehat{\theta}_i^k(\pi^k) = \frac{1}{\tau} \sum_{t=1}^{\tau} \left(\Sigma_i^k\right)^{-1} \phi_i(a_i^{k,t}) r_i^{k,t}$, with the covariance matrix $\Sigma_i^k = \mathbb{E}_{a_i \sim \pi_i^k}\left[\phi_i(a_i)\phi_i(a_i)^\top\right]$. Then, we have the unbiased gradient estimate

$$\widehat{\nabla}_i^k \Phi(a_i) = \left\langle \phi_i(a_i), \widehat{\theta}_i^k(\pi^k) \right\rangle = \frac{1}{\tau} \sum_{t=1}^{\tau} \phi_i(a_i)^\top \left(\Sigma_i^k\right)^{-1} \phi_i(a_i^{k,t}) r_i^{k,t}. \quad (4)$$

*Remark* 4. One difference between Algorithm 3 (decentralized) and Algorithm 1 (centralized) is that in the decentralized algorithm, each player is required to play the same policy for $\tau$ times before an update can be applied. An episode is thus defined for convenience as the time period during which the players' policies are fixed. We make this artificial design mainly for controlling the variance of the gradient estimator $\widehat{\nabla}_i^k \Phi(a_i)$. However, we conjecture that with more careful design and analysis, it should be possible to improve Algorithm 3 so that only one sample is required per episode [Zhang et al., 2020].

**G-optimal Design.** In line 8 and 9, the algorithm performs standard Frank-Wolfe update and mixes the updated policy with an exploration policy $\rho_i$, which is defined as the G-optimal allocation for features $\{\phi_i(a_i)\}_{a_i \in \mathcal{A}_i}$. To be specific, we have

$$\rho_i = \underset{\lambda \in \Delta(\mathcal{A}_i)}{\text{argmin}} \max_{a_i \in \mathcal{A}_i} \|\phi_i(a_i)\|_{\mathbb{E}_{a_i' \sim \lambda}[\phi_i(a_i')\phi_i(a_i')^\top]^{-1}}^2. \quad (5)$$

Here $\rho_i$ guarantees that $\Sigma_i^k$ is invertible and the variance of $\widehat{\nabla}_i^k \Phi(a_i) = \left\langle \phi_i(a_i), \widehat{\theta}_i^k(\pi^k) \right\rangle$ depends only on $F$ instead of the size of action space (Lemma 9) because by the famous Kiefer-Wolfowitz theorem, we have $\max_{a_i \in \mathcal{A}_i} \|\phi_i(a_i)\|_{\mathbb{E}_{a_i' \sim \rho_i}[\phi_i(a_i')\phi_i(a_i')^\top]^{-1}}^2 = F_i \leq F$ [Lattimore and Szepesvári, 2020].

**Frank-Wolfe Update.** Finally, we emphasize that it is crucial to use Frank-Wolfe update because it is compatible with $L_1$ *norm* and we can show that $\Phi$ is $mF$-smooth with respect to the $L_1$ norm (Lemma 11). In contrast, its smoothness for $L_2$ norm will depend on the size of the action space.

Before the game starts, each player $i$ can compute her $\rho_i$ based on her own action set $\mathcal{A}_i$. During the game, all players only have access to their own actions and rewards, which means that Algorithm 3 is fully decentralized. The Nash regret bound for this algorithm is formally stated in Theorem 2 and the proof details are given in Appendix D.1 and D.2.

**Theorem 2.** *Let $T = K\tau$. For congestion game with bandit feedback, by running Algorithm 3 with gradient estimator $\widehat{\nabla}_i^k \Phi$ in (4) and exploration distribution $\rho_i$ in (5), if $K \geq \frac{2F}{m}$, then with probability at least $1 - \delta$, we have*

$$\text{Nash-Regret}(T) := \sum_{k=1}^{K} \tau \max_{i \in [m]} \left( V_i^{\dagger, \pi_{-i}^k} - V_i^{\pi^k} \right) \leq \widetilde{\mathcal{O}} \left( m^2 F^2 T^{5/6} + m^3 F^3 T^{2/3} \right).$$

*For congestion game with semi-bandit feedback, by running Algorithm 3 with gradient estimator $\widetilde{\nabla}_i^k \Phi(a_i)$ and exploration distribution $\tilde{\rho}_i$ defined in Appendix D.3, if $K \geq \frac{2\sqrt{F}}{m}$, then with probability at least $1 - \delta$, we have*

$$\text{Nash-Regret}(T) \leq \widetilde{\mathcal{O}} \left( m^2 F^{3/2} T^{5/6} + m^3 F^2 T^{2/3} \right).$$

# 6 Extension to Independent Markov Congestion Games

In this section, we propose and analyze a Markov extension of the congestion games, called the independent Markov congestion games (IMCGs).

## 6.1 Problem Formulation

**General-sum Markov Games.** A finite-horizon time-inhomogeneous tabular general-sum Markov game is defined by $\mathcal{M} = \{\mathcal{S}, \{\mathcal{A}_i\}_{i=1}^m, H, P, R, s_0\}$, where $\mathcal{S}$ is the state space, $m$ is the number of players, $\mathcal{A}_i$ is the action space of player $i$, $\mathcal{A} = \mathcal{A}_1 \times \cdots \times \mathcal{A}_m$ is the whole action space, $H$ is the time horizon, $s_0$ is the initial state[1], $P = (P_1, P_2, \cdots, P_H)$ with $P_h \in [0,1]^{S \times A \times S}$ as the transition kernel at timestep $h$, $R = \{R_h(\cdot|s_h, \boldsymbol{a}_h)\}_{h=1}^H$ with $R_h(\cdot|s_h, \boldsymbol{a}_h)$ as the reward distribution on $[0, r_{\max}]^m$ with mean $\boldsymbol{r}_h(s_h, a_h) \in [0, r_{\max}]^m$ at timestep $h \in [H]$. At timestep $h$, all players choose their actions simultaneously and a reward vector is sampled $\boldsymbol{r}_h \sim R_h(\cdot|s_h, \boldsymbol{a}_h)$, where $s_h$ is the current state and $\boldsymbol{a}_h = (a_{h,1}, a_{h,2}, \cdots, a_{h,m})$ is the joint action. Each player $i$ receives reward $r_{h,i}$ and the state transits to $s_{h+1} \sim P_h(\cdot|s_h, \boldsymbol{a}_h)$. The objective for each player is to maximize her own total reward. We assume that the initial state $s_1$ is fixed.

A (Markov) policy $\pi$ is a collection of $H$ functions $\{\pi_h : \mathcal{S} \mapsto \Delta(\mathcal{A})\}_{h=1}^H$, each of which maps a state to a distribution over the action space. $\pi$ is a product policy if $\pi_h(\cdot \mid s)$ is a product policy for each $(h, s) \in [H] \times \mathcal{S}$. The value function and $Q$-value function of player $i$ at timestep $h$ under policy $\pi$ are defined as

$$V_{h,i}^{\pi}(s) = \mathbb{E}_\pi \left[ \sum_{h'=h}^{H} r_{h',i}(s_{h'}, \boldsymbol{a}_{h'}) \mid s_h = s \right], \quad Q_{h,i}^{\pi}(s, \boldsymbol{a}) = \mathbb{E}_\pi \left[ \sum_{h'=h}^{H} r_{h',i}(s_{h'}, \boldsymbol{a}_{h'}) \mid s_h = s, \boldsymbol{a}_h = a \right].$$

The best responses and Nash regret can be defined similarly as those for matrix games. In particular, given a policy $\pi$, player $i$'s best response policy is $\pi_{h,i}^{\dagger}(\cdot \mid s) = \text{argmax}_{\mu \in \Delta(\mathcal{A}_i)} V_{h,i}^{\mu, \pi^{-i}}(s)$ and the corresponding value function is denoted as $V_{h,i}^{\dagger, \pi^{-i}}$.

**Definition 3.** *With $\pi^k$ being the policy at $k$th episode, the Nash regret after $K$ episodes is define as*

$$\text{Nash-Regret}(K) = \sum_{k=1}^{K} \max_{i \in [m]} \left( V_{1,i}^{\dagger, \pi_{-i}^k} - V_{1,i}^{\pi^k} \right)(s_1).$$

**Independent Markov Congestion Game.** A general-sum Markov game is an independent Markov congestion game (IMCG) if there exists a facility set $\mathcal{F}$ such that $a_i \subseteq \mathcal{F}$ for any $a_i \in \mathcal{A}_i$, a state space $\mathcal{S} = \prod_{f \in \mathcal{F}} \mathcal{S}^f$, a set of facility reward distributions $\{R_h^f\}_{h \in [H], f \in \mathcal{F}}$ such that if the joint

---

[1]An episode is defined as running $H$ steps from the initial state $s_0$, which is common for the episodic MDP.

action at $s_h$ is $\boldsymbol{a}$, we have $r_{h,i} = \sum_{f \in a_i} r_h^f$, where $r_h^f \sim R_h^f(\cdot|s_h, n^f(\boldsymbol{a}))$ with support on $[0, 1]$ and mean $r_h^f(s_h, n^f(\boldsymbol{a}))$, and a set of transition matrices $\{P_h^f\}_{h \in [H], f \in \mathcal{F}}$ such that $P_h(s'|s, \boldsymbol{a}) = \prod_{f \in \mathcal{F}} P_h^f(s'^f|s^f, n^f(\boldsymbol{a}))$. In other words, at each timestep $h$ and state $s \in \mathcal{S}$, the players are in a congestion game. Meanwhile, each facility has its own state and independent state transition, which only depends on its current state and number of players using that facility. This transition kernel can be viewed as a special case of that in factored MDPs [Szita and Lőrincz, 2009]. The IMCG also admits two types of feedback, semi-bandit feedback and bandit feedback, just like the congestion game. In this paper, we will consider both types of feedback.

## 6.2 Theoretical Guarantee

Summarized in Algorithm 5, our centralized algorithm for IMCGs is naturally extended from the Nash-UCB (Algorithm 1) by incorporating transition kernel estimators, corresponding bonus terms and Bellman backward update. The key idea is to utilize the independent transition structure to remove the dependence on the exponential size of the state space $S = \prod_{f \in \mathcal{F}} S^f$. We tackle this issue by adapting technique from factored MDP [Chen et al., 2020]. The algorithmic details for both types of feedback are deferred into Appendix E. The Nash regret bounds for the two versions of Algorithm 5 are stated in Theorem 3 and the proof details are deferred to Appendix F.

**Theorem 3.** *For independent Markov congestion game with semi-bandit feedback, by running the centralized Algorithm 5, with probability at least $1 - \delta$, we can achieve that*

$$\textit{Nash-Regret}(K) \leq \widetilde{\mathcal{O}} \left( \sum_{f \in \mathcal{F}} F S^f \sqrt{mH^3T} \right) + \widetilde{\mathcal{O}} \left( m^2 H^2 F \sum_{f \neq f'} \left( S^f S^{f'} \right)^2 \right).$$

*Furthermore, if we only have bandit feedback, then by running Algorithm 5 with reward estimator and bonus term in (12) and (13), with probability at least $1 - \delta$, we can achieve that*

$$\textit{Nash-Regret}(K) \leq \widetilde{\mathcal{O}} \left( \sum_{f \in \mathcal{F}} F S^f \sqrt{m^2 H^3 T} \right) + \widetilde{\mathcal{O}} \left( m^2 H^2 F \sum_{f \neq f'} \left( S^f S^{f'} \right)^2 \right).$$

The regret bound in [Liu et al., 2021] is $\widetilde{O}(\sqrt{H^3 S^2 (\prod_{i=1}^{m} A_i)T})$, where both $A_i$ and $S = \prod_{f \in \mathcal{F}} S^f$ can be exponential in $F$. Our bounds have polynomial dependence on all the parameters.

## 7 Conclusion

In this paper, we study sample-efficient learning in congestion games by utilizing the special reward structure. We propose both centralized and decentralized algorithms for congestion games with two types of feedback, all achieving sample complexities only polynomial in the number of facilities. To the best of our knowledge, each one of them is the first sample-efficient learning algorithm for congestion games in its own setting. We further define the independent Markov congestion game (IMCG) as a natural extension of the congestion game into the Markov setting together with a sample-efficient centralized algorithm for both types of feedback.

One promising future direction is to find a sample-efficient decentralized algorithm such that from each player's own perspective, the algorithm is still no-regret. In other words, diminishing regret is guaranteed for the player by running this algorithm even though other players may use policies from different algorithms. Another important future direction is to find sample-efficient centralized/decentralized algorithms that can explicitly find an approximate Nash equilibrium policy.

## Acknowledgements

We sincerely thank Jing Dong for pointing out a mistake in the initial draft of this paper. This work was supported in part by NSF TRIPODS II-DMS 2023166, NSF CCF 2007036, NSF IIS 2110170, NSF DMS 2134106, NSF CCF 2212261, NSF IIS 2143493, NSF CCF 2019844.

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
