# Contents

# A   Additional Motivating Examples

In this section, we present two additional motivating examples of our proposed models.

*Example* 2 (**Web Advertisements**). Consider a set of websites as the facility set and companies who want to advertise their products as the players. Due to budget constraints, each company may only choose some of these websites to put its product ad. For each website, the probability that a user will click on a certain ad (and then buy the product) depends on how many ads are put on the website. If a website receives too many ads, the probability that a user can see a certain ad will decrease, thus making it congested.[2] The reward each company will receive is measured by the amount of products sold during certain period of time, which is bandit feedback.

*Example* 3 (**Server Usage**). Consider a set of servers in a company as the facility set and server users as the players. Each user needs to request several servers to finish her computation task and the cost triggered from each server depends on the number of users requesting that server. Each user will try to minimize the total cost incurred from the servers she requested. As each user can see the cost from all the servers she requested, this is semi-bandit feedback.

# B   Compute $\epsilon$-approximate Nash Equilibrium in Potential Games

In this section, we show that the $\epsilon$-NASH$(\cdot)$ operation in Algorithm 1 can be computed efficiently by using Algorithm 2.

In particular, we first show that the matrix game with reward functions $\overline{Q}_1^k(\cdot), \ldots, \overline{Q}_m^k(\cdot)$ used in Algorithm 1 is a potential game in Lemma 1. Then, we show that Algorithm 2 can efficiently compute an $\epsilon$-approximate Nash equilibrium for potential games and output a product policy as shown in Lemma 2.

**Lemma 1.** *In line 5 of Algorithm 1, the matrix game with reward functions $\overline{Q}_1^k(\cdot), \ldots, \overline{Q}_m^k(\cdot)$ forms a potential game for both settings of semi-bandit feedback and bandit feedback.*

*Proof.* In the setting of semi-bandit feedback, since $\overline{Q}_i^k(\boldsymbol{a}) = \sum_{f \in a_i} (\hat{r}^{k,f} + b^{k,f,\mathrm{r}})(\boldsymbol{a})$, the reward functions $\overline{Q}_1^k(\cdot), \ldots, \overline{Q}_m^k(\cdot)$ form a congestion game, which we know is a potential game [Monderer and Shapley, 1996].

In the setting of bandit feedback, notice that by defining $\tilde{r}^{k,f}(i) = \widehat{\theta}_{i+m(f-1)}^k$ for $(i, f) \in [m] \times \mathcal{F}$, we can have $\tilde{r}_i^k(\boldsymbol{a}) = \left\langle A_i(\boldsymbol{a}), \widehat{\theta}^k \right\rangle = \sum_{f \in a_i} \tilde{r}^{k,f}(n^f(\boldsymbol{a}))$. Therefore, we claim that the desired potential function is

$$\Phi^k(\boldsymbol{a}) = \widetilde{\Phi}^k(\boldsymbol{a}) + \tilde{b}^{k,\mathrm{r}}(\boldsymbol{a}), \quad \text{where} \quad \widetilde{\Phi}^k(\boldsymbol{a}) = \sum_{f \in \mathcal{F}} \sum_{i=1}^{n^f(\boldsymbol{a})} \tilde{r}^{k,f}(i).$$

To see this, by referring to the definition of potential function in congestion game [Monderer and Shapley, 1996], since $\tilde{r}_i^k(\boldsymbol{a}) = \sum_{f \in a_i} \tilde{r}^{k,f}(n^f(\boldsymbol{a}))$, we have that

$$\widetilde{\Phi}^k(a_i, a_{-i}) - \widetilde{\Phi}^k(a_i', a_{-i}) = \tilde{r}_i(a_i, a_{-i}) - \tilde{r}_i(a_i', a_{-i}).$$

As a result, we have

$$\begin{aligned}
&\Phi^k(a_i, a_{-i}) - \Phi^k(a_i', a_{-i}) \\
&= \left( \tilde{r}_i(a_i, a_{-i}) + \tilde{b}^{k,\mathrm{r}}(a_i, a_{-i}) \right) - \left( \tilde{r}_i(a_i', a_{-i}) + \tilde{b}^{k,\mathrm{r}}(a_i', a_{-i}) \right) \\
&= \overline{Q}_i^k(a_i, a_{-i}) - \overline{Q}_i^k(a_i', a_{-i}),
\end{aligned}$$

which means that $\overline{Q}_1^k(\cdot), \ldots, \overline{Q}_m^k(\cdot)$ form a potential game.   $\square$

---

[2]Although the website's intelligent recommendation system may more or less mitigate this effect, it can be considered as a part of the reward function's property.

**Lemma 2.** *Algorithm 2 can output an $\epsilon$-approximate Nash equilibrium.*

*Proof.* Note that if at round $k$, we have $\max_{i \in [m]} \Delta_i \leq \epsilon$, then $\pi^k$ is an $\epsilon$-approximate Nash equilibrium. So we only need to prove that $\max_{i \in [m]} \Delta_i \leq \epsilon$ is satisfied at some round $k \in \{1, \ldots, \lceil \frac{mr_{\max}}{\epsilon} \rceil\}$.

Suppose the potential game $(\{\mathcal{A}_i\}_{i=1}^m, \{r_i\}_{i=1}^m)$ is associated with potential function $\Phi \in [0, \Phi_{\max}]$. Set $\pi^* = \text{argmax}_{\pi \in \prod_{i \in [m]} \Delta(\mathcal{A}_i)} \Phi(\pi)$. Then for any $\pi \in \prod_{i \in [m]} \Delta(\mathcal{A}_i)$, we have

$$\Phi(\pi^*) - \Phi(\pi) = \sum_{i \in [m]} \left( \Phi(\pi_{1:i}^*, \pi_{i+1:m}) - \Phi(\pi_{1:i-1}^*, \pi_{i:m}) \right)$$

$$= \sum_{i \in [m]} \left( V_i^{\pi_{1:i}^*, \pi_{i+1:m}} - V_i^{\pi_{1:i-1}^*, \pi_{i:m}} \right)$$

$$\leq mr_{\max}.$$

As a result, we can set $\Phi_{\max} = mr_{\max}$. On the other hand, if $j = \text{argmax}_{i \in [m]} \Delta_i$ for round $k$, we have

$$\Phi(\pi^{k+1}) - \Phi(\pi^k) = \Phi(\pi_j^{k+1}, \pi_{-j}^k) - \Phi(\pi^k)$$

$$= V_j^{\pi_j^{k+1}, \pi_{-j}^k} - V_j^{\pi^k}$$

$$= r_j(a_j^{k+1}, \pi_{-j}^k) - r_j(\pi^k) \qquad (\pi^k \text{ is deterministic})$$

$$= \Delta_j$$

$$= \max_{i \in [m]} \Delta_i.$$

So there must exist $k \in \{1, \ldots, \lceil \frac{mr_{\max}}{\epsilon} \rceil\}$ such that $\max_{i \in [m]} \Delta_i \leq \epsilon$, otherwise $\Phi(\pi^k)$ increase at least $\epsilon$ at each round, which contradicts $\Phi \in [0, mr_{\max}]$. $\square$

## C  Analysis for Algorithm 1

Recall that the update rule in Algorithm 1 is $\overline{Q}_i^k(\boldsymbol{a}) = \hat{r}_i^k(\boldsymbol{a}) + b_i^{k,\text{r}}(\boldsymbol{a})$, where we have

$$b_i^{k,\text{r}}(\boldsymbol{a}) = \sum_{f \in a_i} b^{k,f,\text{r}}(\boldsymbol{a}), \quad \text{and} \quad b^{k,f,\text{r}}(\boldsymbol{a}) = \sqrt{\frac{\tilde{\iota}}{N^{k,f}(n^f(\boldsymbol{a})) \vee 1}}.$$

For proof convenience, we define auxiliary value functions

$$\underline{Q}_i^k(\boldsymbol{a}) = \hat{r}_i^k(\boldsymbol{a}) - b_i^{k,\text{r}}(\boldsymbol{a}),$$

$$\overline{V}_i^k = \mathbb{E}_{\boldsymbol{a} \sim \pi^k}[\overline{Q}_i^k(\boldsymbol{a})] \quad \text{and} \quad \underline{V}_i^k = \mathbb{E}_{\boldsymbol{a} \sim \pi^k}[\underline{Q}_i^k(\boldsymbol{a})].$$

With these definitions, we now begin to prove Theorem 1.

*Proof of Theorem 1.* **Semi-bandit Feedback.** By the update rules in Algorithm 1, in the setting of semi-bandit feedback, with probability at least $1 - \delta$, simultaneously for all $(k, i, \boldsymbol{a}) \in [K] \times [m] \times \mathcal{A}$, we have

$$\overline{Q}_i^k(\boldsymbol{a}) - r_i(\boldsymbol{a}) = \sum_{f \in a_i} \left[ (\hat{r}^{k,f} - r^f)(\boldsymbol{a}) + b^{k,f,\text{r}}(\boldsymbol{a}) \right] \geq 0.$$

The second inequality above is obtained by using standard Hoeffding's inequality and union bound, Therefore, we have $\overline{Q}_i^k(\boldsymbol{a}) \geq r_i(\boldsymbol{a})$.

Then, since $\pi^k$ is the $\epsilon$-approximate Nash equilibrium policy of $\overline{Q}_1^k, \ldots, \overline{Q}_m^k$, we have

$$\overline{V}_i^k = \mathbb{E}_{\boldsymbol{a} \sim \pi^k}[\overline{Q}_i^k(\boldsymbol{a})] = \max_{\nu \in \Delta(\mathcal{A}_i)} \mathbb{E}_{\boldsymbol{a} \sim (\nu, \pi_{-i}^k)}[\overline{Q}_i^k(\boldsymbol{a})] - \epsilon$$

$$\geq \max_{\nu \in \Delta(\mathcal{A}_i)} \mathbb{E}_{\boldsymbol{a} \sim (\nu, \pi^k_{-i})}[r_i(\boldsymbol{a})] - \epsilon = V_i^{\dagger, \pi^k_{-i}} - \epsilon.$$

Meanwhile, by definition of $\underline{Q}_i^k(\boldsymbol{a})$ and $\underline{V}_i^k$, we can similarly show that $\underline{Q}_i^k(\boldsymbol{a}) \leq r_i(\boldsymbol{a})$ and $\underline{V}_i^k \leq V_i^{\pi^k}$. Therefore, we can have $V_i^{\dagger, \pi^k_{-i}} - V_i^{\pi^k} \leq \overline{V}_i^k - \underline{V}_i^k + \epsilon$.

Now, we define $\widetilde{Q}^k(\boldsymbol{a}) = \max_{i \in [m]} 2b_i^{k,\mathrm{r}}(\boldsymbol{a})$ and $\widetilde{V}^k = \mathbb{E}_{\boldsymbol{a} \sim \pi^k}[\widetilde{Q}^k(\boldsymbol{a})]$. Then, we can notice that

$$\max_{i \in [m]}(\overline{Q}_i^k - \underline{Q}_i^k)(\boldsymbol{a}) \leq \max_{i \in [m]} 2b_i^{k,\mathrm{r}}(\boldsymbol{a}) = \widetilde{Q}^k(\boldsymbol{a}),$$

$$\max_{i \in [m]}(\overline{V}_i^k - \underline{V}_i^k) \leq \mathbb{E}_{\boldsymbol{a} \sim \pi^k}\left[\max_{i \in [m]}(\overline{Q}_i^k - \underline{Q}_i^k)(\boldsymbol{a})\right] \leq \mathbb{E}_{\boldsymbol{a} \sim \pi^k}[\widetilde{Q}^k(\boldsymbol{a})] = \widetilde{V}^k.$$

We further define $\mathcal{M}^k = \mathbb{E}_{\boldsymbol{a} \sim \pi^k}\left[\widetilde{Q}^k(\boldsymbol{a})\right] - \widetilde{Q}^k(\boldsymbol{a}^k) = \widetilde{V}^k - \widetilde{Q}^k(\boldsymbol{a}^k)$. It is not hard to verify that $\mathcal{M}^k$ is a martingale difference sequence with respect to the history from episode 1 to $k - 1$. Meanwhile, since $|b^{k,\mathrm{r}}(\boldsymbol{a})| = \sum_{f \in \mathcal{F}} \sqrt{\frac{\tilde{\iota}}{N^{k,f}(n^f(\boldsymbol{a})) \vee 1}} \leq F\sqrt{\tilde{\iota}}$. Thus, by Azuma-Hoeffding inequality, we have $\sum_{k=1}^K \mathcal{M}^k = \widetilde{\mathcal{O}}\left(F\sqrt{K}\right)$. Therefore, we have

$$
\begin{aligned}
\text{Nash-Regret}(K) &= \sum_{k=1}^K \max_{i \in [m]} \left(V_i^{\dagger, \pi^k_{-i}} - V_i^{\pi^k}\right) \\
&= \sum_{k=1}^K \min\left\{\max_{i \in [m]}\left(V_i^{\dagger, \pi^k_{-i}} - V_i^{\pi^k}\right), F\right\} \\
&\qquad\qquad\qquad \text{(Since the value is always bounded by } F.) \\
&\leq \sum_{k=1}^K \min\left\{\max_{i \in [m]}\left(\overline{V}_i^k - \underline{V}_i^k\right), F\right\} + K\epsilon \\
&\leq \sum_{k=1}^K \min\left\{\widetilde{V}^k, F\right\} + K\epsilon \\
&= \sum_{k=1}^K \left(\min\left\{\widetilde{Q}^k(\boldsymbol{a}^k), F\right\} + \mathcal{M}^k\right) + K\epsilon \\
&\leq \widetilde{\mathcal{O}}\left(F\sqrt{K}\right) + 2\sum_{k=1}^K \left\{\max_{i \in [m]} b_i^{k,\mathrm{r}}(\boldsymbol{a}^k), F\right\} \qquad \text{(By taking } \epsilon = 1/K.) \\
&\leq \widetilde{\mathcal{O}}\left(F\sqrt{K}\right) + 2\sum_{f \in \mathcal{F}}\sum_{k=1}^K \sqrt{\frac{\tilde{\iota}}{N^{k,f}(n^f(\boldsymbol{a}^k)) \vee 1}} \\
&\leq \widetilde{\mathcal{O}}\left(F\sqrt{mK}\right) \qquad\qquad\qquad\qquad\qquad \text{(By Lemma 6.)}
\end{aligned}
$$

**Bandit Feedback.** By using Lemma 3, which guarantees optimistic estimation, we can similarly show that

$$\text{Nash-Regret}(K) \leq \sum_{k=1}^K \mathcal{M}^k + \sum_{k=1}^K \min\left\{2\tilde{b}^{k,\mathrm{r}}(\boldsymbol{a}^k), F\right\} + K\epsilon.$$

To have an upper bound on $\mathcal{M}^k$ here, recall that $\tilde{b}^{k,\mathrm{r}}(\boldsymbol{a}) = \max_{i \in [m]} \|A_i(\boldsymbol{a})\|_{(V^k)^{-1}} \sqrt{\tilde{\beta}_k}$ and $\sqrt{\tilde{\beta}_K} = \widetilde{\mathcal{O}}\left(\sqrt{F\tilde{d}}\right) = \widetilde{\mathcal{O}}(F\sqrt{m})$. Meanwhile, we have $\|A_i(\boldsymbol{a})\|_{(V^k)^{-1}} \leq \|A_i(\boldsymbol{a})\|_I = \|A_i(\boldsymbol{a})\|_2 \leq \sqrt{F}$. Thus, we have $|\mathcal{M}^k| \leq \widetilde{\mathcal{O}}\left(\sqrt{mF^3}\right)$, which by Azuma-Hoeffding inequality implies $\sum_{k=1}^K \mathcal{M}^k = \widetilde{\mathcal{O}}\left(\sqrt{mF^3 K}\right)$.

Then the sum of the bonus terms can be bounded by using Lemma 4. In particular, with $\epsilon = 1/K$, we have

$$\text{Nash-Regret}(K) \leq \widetilde{\mathcal{O}}\left(\sqrt{mF^3 K}\right) + 2\sum_{k=1}^{K} \min\left\{\max_{i\in[m]} \|A_i(\boldsymbol{a}^k)\|_{(V^k)^{-1}} \sqrt{\tilde{\beta}_k}, F\right\}$$

$$\leq \widetilde{\mathcal{O}}\left(\sqrt{mF^3 K}\right) + 2\sqrt{K \sum_{k=1}^{K} \min\left\{\max_{i\in[m]} \|A_i(\boldsymbol{a}^k)\|_{(V^k)^{-1}}^2 \tilde{\beta}_k, F^2\right\}}$$

$$\leq \widetilde{\mathcal{O}}\left(\sqrt{mF^3 K}\right) + \sqrt{\widetilde{\mathcal{O}}\left(mF^2 K\right) \sum_{k=1}^{K} \min\left\{\max_{i\in[m]} \|A_i(\boldsymbol{a}^k)\|_{(V^k)^{-1}}^2, 1\right\}}$$

$$\text{(Since } \tilde{\beta}_k = \widetilde{\mathcal{O}}\left(mF^2\right).)$$

$$\leq \widetilde{\mathcal{O}}\left(\sqrt{mF^3 K}\right) + \widetilde{\mathcal{O}}\left(\sqrt{mF^2 K \cdot mF}\right) \qquad \text{(By Lemma 4.)}$$

$$\leq \widetilde{\mathcal{O}}\left(mF^{3/2}\sqrt{K}\right).$$

$\square$

## C.1 Lemmas for Bandit Feedback

The following lemma, as a direct corollary of the confidence bound for least square estimators, shows that the reward estimation error can be bounded by the reward bonus term.

**Lemma 3.** *With probability at least $1-\delta$, simultaneously for all $(i, k, \boldsymbol{a})$, it holds that $|(\tilde{r}_i^k - r_i)(\boldsymbol{a})| \leq \tilde{b}^{k,\mathrm{r}}(\boldsymbol{a})$, where $\tilde{r}_i^k$ and $\hat{b}^{k,\mathrm{r}}$ are defined in (2).*

*Proof.* By construction, we have

$$|(\tilde{r}_i^k - r_i)(\boldsymbol{a})| = \left|\left\langle A_i(\boldsymbol{a}), \hat{\theta} - \theta \right\rangle\right|$$

$$\leq \|A_i(\boldsymbol{a})\|_{(V^k)^{-1}} \left\|\hat{\theta} - \theta\right\|_{V^k}$$

$$\overset{(i)}{\leq} \|A_i(\boldsymbol{a})\|_{(V^k)^{-1}} \left(\|\theta\|_2 + \sqrt{F\log\left(\det(V^k)\right) + F\tilde{\iota}}\right),$$

where the inequality (i) above holds because of Theorem 20.5 in Lattimore and Szepesvári [2020] and the fact that the reward noise is $\sqrt{F}$-subGaussian. Since each element in $\theta$ is bounded in $[0, 1]$ by construction, we have $\|\theta\|_2 \leq \sqrt{\tilde{d}}$.

Then, by Lemma 4, we have $\det\left(V^k\right) \leq \left(1 + \frac{mkF}{\tilde{d}}\right)^{\tilde{d}}$ since by construction $\|A_i(\boldsymbol{a})\|_2^2 \leq F$.

Finally, to make this bound valid for all player $i \in [m]$, we only need to take maximization over $i \in [m]$. Therefore, with probability at least $1 - \delta$, we have

$$|(\tilde{r}_i^k - r_i)(\boldsymbol{a})| \leq \max_{i\in[m]} \|A_i(\boldsymbol{a})\|_{(V^k)^{-1}} \sqrt{\tilde{\beta}_k} = \tilde{b}^{k,\mathrm{r}}(\boldsymbol{a}),$$

where $\sqrt{\tilde{\beta}_k} = \sqrt{\tilde{d}} + \sqrt{F\tilde{d}\log\left(1 + \frac{mkF}{\tilde{d}}\right) + F\tilde{\iota}}$. $\square$

The following is a variant of the famous elliptical potential lemma, which helps bound the sum of reward bonus under bandit feedback. Here, we apply some techniques from the proof of Lemma 19.4 in Lattimore and Szepesvári [2020].

**Lemma 4.** *Let $K, m \geq 1$ be integers. Suppose $V^k = I + \sum_{k'=1}^{k-1}\sum_{i=1}^{m} A_i^{k'}\left(A_i^{k'}\right)^{\top}$, where $A_i^{k'} \in \mathbb{R}^d$ and $\left\|A_i^{k'}\right\|_2^2 \leq F$. Then, it holds that*

$$\det\left(V^k\right) \leq \left(1 + \frac{mkF}{d}\right)^d, \quad \text{and} \quad \sum_{k=1}^{K} \min\left\{\max_{i\in[m]} \|A_i^k\|_{(V^k)^{-1}}^2, 1\right\} \leq 2d\log\left(1 + \frac{mKF}{d}\right).$$

*Proof.* For the first upper bound about $\det\left(V^k\right)$, we have

$$\det\left(V^k\right) = \prod_{j=1}^{d} \lambda_j \qquad\qquad (\lambda_1, \dots, \lambda_d \text{ are eigenvalues of } V^k)$$

$$\leq \left(\frac{\operatorname{tr}\left(V^k\right)}{d}\right)^d \qquad\qquad \text{(By AM-GM inequality)}$$

$$= \left(\frac{\operatorname{tr}\left(I\right) + \sum_{k'=1}^{k-1}\sum_{i=1}^{m}\left\|A_i^{k'}\right\|_2^2}{d}\right)^d$$

$$\leq \left(1 + \frac{mkF}{d}\right)^d. \qquad\qquad \left(\text{Since } \left\|A_i^{k'}\right\|_2^2 \leq F.\right)$$

For the second upper bound. First, we notice that $\min\{1, x\} \leq 2\log(1+x)$ for any $x \geq 0$. Thus, we have

$$\sum_{k=1}^{K}\min\left\{1, \max_{i\in[m]}\left\|A_i^k\right\|_{(V^k)^{-1}}^2\right\} \leq 2\sum_{k=1}^{K}\log\left(1 + \max_{i\in[m]}\left\|A_i^k\right\|_{(V^k)^{-1}}^2\right).$$

Then, for $k \geq 2$, we can notice that

$$V^k = V^{k-1} + \sum_{i=1}^{m}A_i^{k-1}\left(A_i^{k-1}\right)^\top$$

$$= \left(V^{k-1}\right)^{1/2}\left(I + \left(V^{k-1}\right)^{-1/2}\left(\sum_{i=1}^{m}A_i^{k-1}\left(A_i^{k-1}\right)^\top\right)\left(V^{k-1}\right)^{-1/2}\right)\left(V^{k-1}\right)^{1/2}$$

$$= \left(V^{k-1}\right)^{1/2}\left(I + \sum_{i=1}^{m}\left(\left(V^{k-1}\right)^{-1/2}A_i^{k-1}\right)\left(\left(V^{k-1}\right)^{-1/2}A_i^{k-1}\right)^\top\right)\left(V^{k-1}\right)^{1/2}.$$

Therefore, we have

$$\det\left(V^k\right) = \det\left(V^{k-1}\right)\det\left(I + \sum_{i=1}^{m}\left(\left(V^{k-1}\right)^{-1/2}A_i^{k-1}\right)\left(\left(V^{k-1}\right)^{-1/2}A_i^{k-1}\right)^\top\right)$$

$$\geq \det\left(V^{k-1}\right)\left(1 + \max_{i\in[m]}\left\|A_i^{k-1}\right\|_{(V^{k-1})^{-1}}^2\right) \qquad\qquad \text{(By Lemma 5.)}$$

$$\geq \prod_{k'=1}^{k-1}\left(1 + \max_{i\in[m]}\left\|A_i^{k'}\right\|_{(V^{k'})^{-1}}^2\right). \qquad\qquad \left(\text{Since by definition, } V^1 = I.\right)$$

As a result, we have

$$\sum_{k=1}^{K}\min\left\{\max_{i\in[m]}\left\|A_i^k\right\|_{(V^k)^{-1}}^2, 1\right\} \leq 2\sum_{k=1}^{K}\log\left(1 + \max_{i\in[m]}\left\|A_i^k\right\|_{(V^k)^{-1}}^2\right)$$

$$\leq 2\log\left(\det\left(V^{K+1}\right)\right)$$

$$\leq 2d\log\left(1 + \frac{mKF}{d}\right).$$

$\square$

## C.2 Technical Lemmas

**Lemma 5.** *Let $y_1, \dots, y_m \in \mathbb{R}^d$ be a set of vectors. Then, it holds that*

$$\det\left(I + \sum_{i=1}^{m}y_i y_i^\top\right) \geq 1 + \max_{i\in[m]}\|y_i\|_2^2.$$

*Proof.* Since $I + \sum_{i=1}^m y_i y_i^\top \succeq I + y_i y_i^\top$ for any $i \in [m]$, we have $\det\left(I + \sum_{i=1}^m y_i y_i^\top\right) \geq \det\left(I + y_i y_i^\top\right)$ for any $i \in [m]$. That is, we have

$$\det\left(I + \sum_{i=1}^m y_i y_i^\top\right) \geq \max_{i \in [m]} \det\left(I + y_i y_i^\top\right) = 1 + \max_{i \in [m]} \|y_i\|_2^2.$$

The last line above holds because the matrix $I + y_i y_i^\top$ has eigenvalues $1 + \|y_i\|_2^2$ and 1. $\quad\square$

**Lemma 6.** *For any $f \in \mathcal{F}$, it holds that*

$$\sum_{k=1}^K \sqrt{\frac{1}{\mathcal{N}^{k,f}(n^f(\boldsymbol{a}^k)) \vee 1}} \leq \tilde{\mathcal{O}}\left(\sqrt{mK}\right).$$

*Proof.* Here, we have

$$
\begin{aligned}
\sum_{k=1}^K \sqrt{\frac{1}{N^{k,f}(n^f(\boldsymbol{a}^k)) \vee 1}} &= \sum_{n=0}^m \sum_{\ell=1}^{N^{K,f}(n)} \sqrt{\frac{1}{\ell}} \\
&\leq 2 \sum_{n=0}^m \sqrt{N^{K,f}(n)} \qquad \text{(By standard technique)} \\
&\leq 2 \sqrt{(m+1) \sum_{n=0}^m N^{K,f}(n)} \\
&= \tilde{\mathcal{O}}\left(\sqrt{mK}\right).
\end{aligned}
$$

The last equality above is based on a pigeon-hold principle argument similar to Lemma 20. $\quad\square$

# D  Analysis for Algorithm 3

## D.1  Exploration Distribution and Smoothness

We choose the exploration distribution to be the G-optimal design and we have the following properties.

**Lemma 7.** *(Unbiasedness) For any episode $k \in [K]$, $i \in [m]$ and $a \in \mathcal{A}_i$, we have*

$$\mathbb{E}_k\left[\widehat{\nabla}_i^k \Phi(a)\right] = \nabla_i^k \Phi(a),$$

*where $\mathbb{E}_k[\cdot]$ is taken over all the randomness before episode $k$.*

*Proof.* By the definition of $\widehat{\nabla}_i^k \Phi(a)$, we have

$$
\begin{aligned}
\mathbb{E}_k\left[\widehat{\nabla}_i^k \Phi(a)\right] &= \mathbb{E}_k\left\langle \phi_i(a), \widehat{\theta}_i^k(\pi^k)\right\rangle \\
&= \mathbb{E}_k\left[\frac{1}{\tau} \sum_{t=1}^\tau \phi_i(a)^\top [\Sigma_i^k]^{-1} \phi_i(a_i^{k,t}) r_i^{k,t}\right] \\
&= \mathbb{E}_k\left[\phi_i(a)^\top [\Sigma_i^k]^{-1} \phi_i(a_i^{k,1}) r_i^{k,1}\right] \\
&= \mathbb{E}_k\left[\phi_i(a)^\top [\Sigma_i^k]^{-1} \phi_i(a_i^{k,1}) \phi_i(a_i^{k,1})^\top \theta_i^{k,1}(\pi^k)\right] \\
&= \sum_{a_i^k \in \mathcal{A}_i} \pi_i^k(a_i^{k,1}) \phi_i^\top(a) [\Sigma_i^k]^{-1} \phi_i(a_i^{k,1}) \phi_i(a_i^{k,1})^\top \theta_i(\pi^k) \\
&\qquad (a_i^{k,1} \text{ only depends on } \pi_i^k \text{ and } \theta_i^{k,1}(\pi^k) \text{ only depends on } \pi_{-i}^k) \\
&= \phi_i^\top(a) [\Sigma_i^k]^{-1} \left[\sum_{a_i^k \in \mathcal{A}_i} \pi_i^k(a_i^{k,1}) \phi_i(a_i^{k,1}) \phi_i(a_i^{k,1})^\top\right] \theta_i(\pi^k)
\end{aligned}
$$

$$=\phi_i^\top(a_i)\theta_i(\pi^k)$$
$$=\nabla_i^k\Phi(a).$$

$\square$

**Lemma 8.** *For any episode $k \in [K]$, $i \in [m]$ and $a \in \mathcal{A}_i$, we have*

$$\left|\phi_i(a)^\top[\Sigma_i^k]^{-1}\phi_i(a_i^{k,t})r_i^{k,t}\right| \leq \frac{F^2}{\gamma}.$$

*Proof.* As $\pi_i^k = (1-\gamma)(\nu\widetilde{\pi}_i^k + (1-\gamma)\pi_i^{k-1}) + \gamma\rho_i$, we have

$$\Sigma_i^k = \mathbb{E}_{a_i \sim \pi_i^k}\phi_i(a_i)\phi_i(a_i)^\top \succeq \gamma\mathbb{E}_{a_i \sim \rho_i}\phi_i(a_i)\phi_i(a_i)^\top,$$

and $\rho_i$ is the G-optimal design with respect to $\phi_i(\cdot)$, for any action $a \in \mathcal{A}_i$ we have

$$\|\phi_i(a)\|_{[\Sigma_i^k]^{-1}}^2 \leq \frac{1}{\gamma}\|\phi_i(a)\|_{[\mathbb{E}_{a_i \sim \rho_i}\phi_i(a_i)\phi_i(a_i)^\top]^{-1}}^2 \leq \frac{F}{\gamma}.$$

Then for any $t \in [\tau]$, since $|r_i^{k,t}| \leq F$, we have

$$\left|r_i^{k,t}\phi_i^\top(a)[\Sigma_i^k]^{-1}\phi_i(a_i^{k,t})\right| \leq \left|r_i^{k,t}\right|\|\phi_i(a)\|_{[\Sigma_i^k]^{-1}}\left\|\phi_i(a_i^{k,t})\right\|_{[\Sigma_i^k]^{-1}} \leq \frac{F^2}{\gamma}.$$

As a result, we have

$$\left|\widehat{\nabla}_i^k\Phi(a)\right| = \left|\frac{1}{\tau}\sum_{t=1}^\tau \phi_i(a)^\top[\Sigma_i^k]^{-1}\phi_i(a_i^{k,t})r_i^{k,t}\right| \leq \frac{F^2}{\gamma}$$

$\square$

**Lemma 9.** *For any episode $k \in [K]$, $i \in [m]$ and $a \in \mathcal{A}_i$, we have*

$$\mathbb{E}_k\left[\left(\phi_i(a)^\top[\Sigma_i^k]^{-1}\phi_i(a_i^{k,t})r_i^{k,t}\right)^2\right] \leq \frac{F^3}{\gamma}.$$

*Proof.* We first show that for any $t \in [\tau]$, we have

$$\mathbb{E}_k\left[\left(\phi_i(a)^\top[\Sigma_i^k]^{-1}\phi_i(a_i^{k,t})r_i^{k,t}\right)^2\right]$$

$$\leq F^2\mathbb{E}_k\left[\left(\phi_i(a)^\top[\Sigma_i^k]^{-1}\phi_i(a_i^{k,t})\right)^2\right]$$

$$\leq F^2\mathbb{E}_k\left[\phi_i(a)^\top[\Sigma_i^k]^{-1}\phi_i(a_i^{k,t})\phi_i(a_i^{k,t})^\top[\Sigma_i^k]^{-1}\phi_i(a)^\top\right]$$

$$= F^2\phi_i(a)^\top[\Sigma_i^k]^{-1}\phi_i(a)$$

$$\leq \frac{F^3}{\gamma}.$$

$\square$

**Lemma 10.** *With probability $1 - \delta$, for all $k \in [K]$, $i \in [m]$ and $a \in \mathcal{A}_i$, we have*

$$\left|\widehat{\nabla}_i^k\Phi(a) - \nabla_i^k\Phi(a)\right| \leq c\sqrt{\frac{F^4\log(mK/\delta)}{\gamma\tau}} + \frac{cF^3\log(mK/\delta)}{\gamma\tau}$$

*Proof.* Recall that

$$\widehat{\nabla}_i^k\Phi(a_i) = \frac{1}{\tau}\sum_{t=1}^\tau \phi_i^\top(a_i)[\Sigma_i^k]^{-1}r_i^{k,t}\phi_i(a_i^{k,t}),$$

and $(a_i^{k,t}, r_i^{k,t})$ are drawn independently at each $t \in [\tau]$. Lemma 7 shows that $\widehat{\nabla}_i^k \Phi(a_i)$ is an unbiased estimate of $\nabla_i^k \Phi(a_i)$ In addition, Lemma 8 shows that $\phi_i^\top(a_i)[\Sigma_i^k]^{-1} r_i^{k,t} \phi_i(a_i^{k,t})$ is bounded by $F^2/\gamma$ and Lemma 9 shows that its second moment is bounded by $F^3/\gamma$. Then by Bernstein's inequality, for a fixed $k \in [K]$, $i \in [m]$ and $a \in \mathcal{A}_i$, with probability $1 - \delta$, we have

$$\left| \widehat{\nabla}_i^k \Phi(a) - \nabla_i^k \Phi(a) \right| \leq \sqrt{\frac{2F^3 \log(2/\delta)}{\gamma \tau}} + \frac{3F^2 \log(2/\delta)}{2\gamma \tau}.$$

The argument holds by applying the union bound and the fact that $|\mathcal{A}_i| \leq 2^F$.

$\square$

**Lemma 11.** $\Phi(\cdot)$ is $mF$-Lipschitz and $mF$-smooth with respect to the L1 norm $\|\cdot\|_1$.

*Proof.* Recall that $\Phi(\pi) = \mathbb{E}_{\boldsymbol{a} \sim \pi} \Phi(\boldsymbol{a})$ and $\Phi(\boldsymbol{a}) \in [0, mF]$.

$$\begin{aligned}
\Phi(\pi) - \Phi(\pi') &= \mathbb{E}_{\boldsymbol{a} \sim \pi} \Phi(\boldsymbol{a}) - \mathbb{E}_{\boldsymbol{a} \sim \pi'} \Phi(\boldsymbol{a}) \\
&= \sum_{i \in [m]} \mathbb{E}_{a_{1:i-1} \sim \pi'_{1:i-1}, a_{i:m} \sim \pi_{i:m}} \Phi(\boldsymbol{a}) - \mathbb{E}_{a_{1:i} \sim \pi'_{1:i}, a_{i+1:m} \sim \pi_{i+1:m}} \Phi(\boldsymbol{a}) \\
&\leq \sum_{i \in [m]} \|\pi_i - \pi'_i\|_1 \cdot \|\Phi\|_\infty \\
&\leq mF \|\pi - \pi'\|_1.
\end{aligned}$$

Similarly we have $\nabla_\pi \Phi(a_i) = \mathbb{E}_{a_{-i} \sim \pi_{-i}} \Phi(a_i, a_{-i})$. As a result, we have

$$\|\nabla_\pi \Phi - \nabla_{\pi'} \Phi\|_\infty \leq mF \|\pi - \pi'\|_1.$$

$\square$

**Definition 4.** (Frank Wolfe Gap) The Frank Wolfe gap of a joint strategy $\pi$ for $\Phi(\cdot)$ is defined as

$$G(\pi) = \max_{\pi'} \langle \pi' - \pi, \nabla_\pi \Phi \rangle.$$

**Lemma 12.** *Suppose the Frank Wolfe gap of $\pi$ is $\epsilon$. Then $\pi$ is an $\epsilon$-Nash policy.*

*Proof.* For a fixed player $i$, suppose player $i$ change her strategy to $\pi'_i$.

$$\begin{aligned}
V_i^{\pi'_i, \pi_{-i}} - V_i^\pi &= \Phi(\pi'_i, \pi_{-i}) - \Phi(\pi) \\
&= \langle \pi'_i - \pi_i, \nabla_{\pi_i} \Phi \rangle \\
&\leq \max_{\pi'} \langle \pi' - \pi, \nabla_\pi \Phi \rangle \\
&\leq \epsilon.
\end{aligned}$$

$\square$

### D.2 Analysis for Frank Wolfe in Bandit Feedback

**Theorem 4.** *Let $T = K\tau$. For the congestion game with bandit feedback, by running Algorithm 3 with gradient estimator $\widehat{\nabla}_i^k \Phi$ in (4) and exploration distribution $\rho_i$ in (5), setting parameters $\nu = \frac{F}{m\sqrt{K}}$, $\gamma = \frac{F}{mK}$ and $\tau = K^2$, if $K \geq \frac{2F}{m}$, then with probability $1 - \delta$, we have*

$$Nash\text{-}Regret(T) = \tau \sum_{k=1}^K G(\pi^k) = \widetilde{\mathcal{O}}\left(m^2 F^2 T^{5/6} + m^3 F^3 T^{2/3}\right).$$

*Proof.* Set $\nabla^k \Phi = \nabla \Phi(\Pi^k) \in \mathbb{R}^A$ and $\nabla_i^k \Phi = \nabla^k \Phi(\pi_i) \in \mathbb{R}^{A_i}$. As we have $\Phi(\cdot)$ is $mF$-smooth w.r.t. $\|\cdot\|_1$, we have

$$\Phi(\pi^{k+1}) \geq \Phi(\pi^k) + \langle \nabla \Phi(\pi^k), \pi^{k+1} - \pi^k \rangle - \frac{mF}{2} \|\pi^{k+1} - \pi^k\|_1^2$$

$$
\begin{aligned}
=&\Phi(\pi^k) + (1-\gamma)\nu\left\langle\nabla\Phi(\pi^k), \widetilde{\pi}^{k+1} - \pi^k\right\rangle + \gamma\left\langle\nabla^k\Phi, \rho - \pi^k\right\rangle \\
&- \frac{mF}{2}\left(2\nu^2\left\|\widetilde{\pi}^k - \pi^k\right\|_1^2 + 2\gamma^2\left\|\rho - \pi^k\right\|_1^2\right) \\
\geq&\Phi(\pi^k) + (1-\gamma)\nu\left\langle\nabla\Phi(\pi^k), \widetilde{\pi}^{k+1} - \pi^k\right\rangle - \gamma\left\|\nabla^k\Phi\right\|_\infty\left\|\rho - \pi^k\right\|_1 \\
&- \frac{mF}{2}\left(2\nu^2\left\|\widetilde{\pi}^k - \pi^k\right\|_1^2 + 2\gamma^2\left\|\rho - \pi^k\right\|_1^2\right) \\
\geq&\Phi(\pi^k) + (1-\gamma)\nu\left\langle\nabla\Phi(\pi^k), \widetilde{\pi}^{k+1} - \pi^k\right\rangle - 2\gamma m^2 F - 4m^3 F(\nu^2 + \gamma^2).
\end{aligned}
$$
$$\text{(By Lemma 11.)}$$

Define the true target policy at episode $k$

$$
\widehat{\pi}_i^{k+1} = \operatorname*{argmax}_{\pi_i}\left\langle\pi_i, \nabla_i\Phi(\pi_i^k)\right\rangle,
$$

and the Frank Wolfe gap of joint strategy $\pi$

$$
G(\pi) = \max_{\pi'}\left\langle\pi' - \pi, \nabla\Phi(\pi)\right\rangle.
$$

Then we have

$$
\begin{aligned}
\left\langle\nabla\Phi(\pi^k), \widetilde{\pi}^{k+1} - \pi^k\right\rangle =&\left\langle\widehat{\nabla}^k\Phi(\pi^k), \widetilde{\pi}^{k+1} - \pi^k\right\rangle + \left\langle\nabla\Phi(\pi^k) - \widehat{\nabla}^k\Phi(\pi^k), \widetilde{\pi}^{k+1} - \pi^k\right\rangle \\
\geq&\left\langle\widehat{\nabla}^k\Phi(\pi^k), \widehat{\pi}^{k+1} - \pi^k\right\rangle + \left\langle\nabla\Phi(\pi^k) - \widehat{\nabla}^k\Phi(\pi^k), \widetilde{\pi}^{k+1} - \pi^k\right\rangle \\
=&\left\langle\nabla\Phi(\pi^k), \widehat{\pi}^{k+1} - \pi^k\right\rangle + \left\langle\nabla\Phi(\pi^k) - \widehat{\nabla}^k\Phi(\pi^k), \widetilde{\pi}^{k+1} - \widehat{\pi}^{k+1}\right\rangle \\
\geq& G(\pi^k) - 2m\left\|\nabla\Phi(\pi^k) - \widehat{\nabla}^k\Phi(\pi^k)\right\|_\infty \\
\geq& G(\pi^k) - c\sqrt{\frac{m^2 F^4\log(mK/\delta)}{\gamma\tau}} - \frac{cmF^3\log(mK/\delta)}{\gamma\tau}
\end{aligned}
$$

Apply it to the previous bound and we have

$$
\begin{aligned}
\Phi(\pi^{k+1}) \geq&\Phi(\pi^k) + (1-\gamma)\nu G(\pi^k) - c\frac{(1-\gamma)\nu}{\sqrt{\gamma\tau}}\sqrt{m^2 F^4\log(mK/\delta)} \\
&- c\frac{(1-\gamma)\nu}{\gamma\tau}mF^3\log(mK/\delta) - \gamma 2m^2 F - 4m^3 F(\nu^2 + \gamma^2).
\end{aligned}
$$

Summing over $k \in [K]$ and we get

$$
\begin{aligned}
\sum_{k=1}^{K} G(\pi^k) \leq&\frac{\Phi(\pi^{K+1}) - \Phi(\pi^1)}{(1-\gamma)\nu} + c\frac{K}{\sqrt{\gamma\tau}}\sqrt{m^2 F^4\log(mK/\delta)} + c\frac{K}{\gamma\tau}mF^3\log(mK/\delta) \\
&+ \frac{2m^2 FK\gamma}{(1-\gamma)\nu} + \frac{4(\nu^2 + \gamma^2)m^3 FK}{(1-\gamma)\nu}.
\end{aligned}
$$

Set $\nu = \frac{F}{m\sqrt{K}}, \gamma = \frac{F}{mK}, \tau = K^2$ and notice that when $K \geq \frac{2F}{m}$, we have $1 - \gamma \geq \frac{1}{2}$. Since $\Phi(\cdot)$ is bounded in $[0, mF]$, we can have

$$
\sum_{k=1}^{K} G(\pi^k) = \widetilde{\mathcal{O}}\left(m^2 F^2 K^{1/2} + m^3 F^3\right).
$$

Then by Lemma 12, for $T = K\tau$, we have

$$
\text{Nash-Regret}(T) = \tau\sum_{k=1}^{K} G(\pi^k) = \widetilde{\mathcal{O}}\left(m^2 F^2 T^{5/6} + m^3 F^3 T^{2/3}\right).
$$

$\square$

### D.3 Algorithm and Analysis for Semi-bandit Feedback

In the setting of semi-bandit feedback, we will need a different gradient estimator $\widetilde{\nabla}_i^k \Phi(a_i)$ and a different exploration distribution $\tilde{\rho}_i$ to utilize the extra reward information from each chosen facility.

Based on the analysis in Section 5, using (3), we have $\nabla_i^k \Phi(a_i) = \sum_{f \in a_i} [\theta_i(\pi^k)]_f$, where $[\theta_i(\pi^k)]_f = \mathbb{E}_{a_{-i} \sim \pi_{-i}^k} \left[ r^f(n^f(a_{-i}) + 1) \right]$. Meanwhile, in semi-bandit feedback, the mean of $t$-th reward player $i$ received for facility $f$ at episode $k$ is $r^f(n^f(a_i^{k,t}, a_{-i}^{k,t}))$. Therefore, we can use inverse propensity score (IPS) estimator to estimate $[\theta_i(\pi^k)]_f$. In particular, we have

$$[\widetilde{\theta}_i^k(\pi^k)]_f = \frac{1}{\tau} \sum_{t=1}^{\tau} [\widetilde{\theta}_i^{k,t}(\pi^k)]_f, \quad \text{where} \quad [\widetilde{\theta}_i^{k,t}(\pi^k)]_f = \frac{r^{k,t,f} \mathbb{1}\left\{ f \in a_i^{k,t} \right\}}{\mathbb{P}_{a_i \sim \pi_i^k}(f \in a_i)}.$$

Then, we can naturally have

$$\widetilde{\nabla}_i^k \Phi(a_i) = \sum_{f \in a_i} [\widetilde{\theta}_i^k(\pi^k)]_f. \tag{6}$$

Furthermore, by Lemma 14, we can see that by using $\tilde{\rho}_i$ computed by Algorithm 4, for all players, we have $\mathbb{P}_{a_i \sim \pi_i^k}(f \in a_i) \geq \frac{\gamma}{2F}$ for all $f \in \bigcup_{a_i \in \mathcal{A}_i} a_i$.

Properties of the IPS estimator are summarized in Lemma 15. By using these properties, we can have the following lemma.

**Lemma 13.** *With probability $1 - \delta$, for all $k \in [K]$, $i \in [m]$ and $a_i \in \mathcal{A}_i$, we have*

$$\left| \widetilde{\nabla}_i^k \Phi(a_i) - \nabla_i^k \Phi(a_i) \right| \leq \sqrt{\frac{4F^3 \log(2mFK/\delta)}{\gamma\tau}} + \frac{2F^2 \log(2mFK/\delta)}{\gamma\tau}.$$

*Proof.* By Lemma 15 and Bernstein's inequality, simultaneously for all $(i, k, f) \in [m] \times [K] \times \mathcal{F}$, with probability at least $1 - \delta$, we have

$$\left| [\widetilde{\theta}_i^k(\pi^k)]_f - [\theta_i(\pi^k)]_f \right| \leq \sqrt{\frac{4F \log(2mFK/\delta)}{\gamma\tau}} + \frac{2F \log(2mFK/\delta)}{\gamma\tau}.$$

Since $\widetilde{\nabla}_i^k \Phi(a_i) = \sum_{f \in a_i} [\widetilde{\theta}_i^k(\pi^k)]_f$, by triangle inequality, we have

$$\left| \widetilde{\nabla}_i^k \Phi(a_i) - \nabla_i^k \Phi(a_i) \right| \leq \sqrt{\frac{4F^3 \log(2mFK/\delta)}{\gamma\tau}} + \frac{2F^2 \log(2mFK/\delta)}{\gamma\tau}.$$

$\square$

With this more refined gradient estimator, we can now have the following theorem.

**Theorem 5.** *Let $T = K\tau$. For the congestion game with semi-bandit feedback, by running Algorithm 3 with gradient estimator $\widetilde{\nabla}_i^k \Phi$ in (6) and exploration distribution $\tilde{\rho}_i$ in Algorithm 4, setting parameters $\nu = \frac{\sqrt{F}}{m\sqrt{K}}$, $\gamma = \frac{\sqrt{F}}{mK}$ and $\tau = K^2$, if $K \geq \frac{2\sqrt{F}}{m}$, then with probability $1 - \delta$, we have*

$$\textit{Nash-Regret}(T) = \tau \sum_{k=1}^{K} G(\pi^k) = \widetilde{\mathcal{O}} \left( m^2 F^{3/2} T^{5/6} + m^3 F^2 T^{2/3} \right).$$

*Proof.* By following the proof of Theorem 4 and applying the concentration inequality in Lemma 13, we can have

$$\Phi(\pi^{k+1}) \geq \Phi(\pi^k) + (1 - \gamma)\nu G(\pi^k) - \frac{(1 - \gamma)\nu}{\sqrt{\gamma\tau}} \sqrt{4m^2 F^3 \log(2mK/\delta)}$$

$$- \frac{2(1 - \gamma)\nu}{\gamma\tau} mF^2 \log(mK/\delta) - \gamma 2m^2 F - 4m^3 F(\nu^2 + \gamma^2).$$

Summing over $k \in [K]$ and we get

$$\sum_{k=1}^{K} G(\pi^k) \leq \frac{\Phi(\pi^{K+1}) - \Phi(\pi^1)}{(1-\gamma)\nu} + \frac{K}{\sqrt{\gamma\tau}}\sqrt{4m^2 F^3 \log(mK/\delta)} + \frac{2K}{\gamma\tau}mF^2 \log(mK/\delta)$$

$$+ \frac{2m^2 FK\gamma}{(1-\gamma)\nu} + \frac{4(\nu^2 + \gamma^2)m^3 FK}{(1-\gamma)\nu}.$$

Set $\nu = \frac{\sqrt{F}}{m\sqrt{K}}$, $\gamma = \frac{\sqrt{F}}{mK}$, $\tau = K^2$ and notice that when $K \geq \frac{2\sqrt{F}}{m}$, we have $1 - \gamma \geq \frac{1}{2}$. Thus, we can have

$$\sum_{k=1}^{K} G(\pi^k) = \widetilde{\mathcal{O}}\left(m^2 F^{3/2} K^{1/2} + m^3 F^2\right).$$

Then by Lemma 12, for $T = K\tau$, we have

$$\text{Nash-Regret}(T) = \tau \sum_{k=1}^{K} G(\pi^k) = \widetilde{\mathcal{O}}\left(m^2 F^{3/2} T^{5/6} + m^3 F^2 T^{2/3}\right).$$

$\square$

## D.4 Lemmas for Semi-bandit Feedback

---
**Algorithm 4** Compute Exploration Distribution $\tilde{\rho}_i$

---
1: **Input:** $\mathcal{A}_i$, player $i$-th action set
2: Initialize $\widetilde{\mathcal{A}}_i \leftarrow \emptyset$
3: **for** $a_i$ in $\mathcal{A}_i$ **do**
4:   **if** $\exists f \in a_i$ such that $f \notin \bigcup_{a'_i \in \widetilde{\mathcal{A}}_i} a'_i$ **then**
5:     $\widetilde{\mathcal{A}}_i \leftarrow \widetilde{\mathcal{A}}_i \cup \{a_i\}$
6:   **if** $\mathcal{F}_i = \bigcup_{a'_i \in \widetilde{\mathcal{A}}_i} a'_i$ **then**
7:     **break**
8: Assign $\tilde{\rho}_i(a_i) \leftarrow \frac{1}{2F}$ for each $a_i \in \widetilde{\mathcal{A}}_i$
9: Assign remaining probability mass arbitrarily to actions in $\mathcal{A} \setminus \widetilde{\mathcal{A}}_i$
10: **return** $\tilde{\rho}_i$

---

**Lemma 14.** *Let $\mathcal{F}_i = \bigcup_{a_i \in \mathcal{A}_i} a_i$. For any player $i$, if $\tilde{\rho}_i$ is the output of Algorithm 4 and $\pi_i^k$ contains a mixture of $\tilde{\rho}_i$ with weight $\gamma$, then we have $\mathbb{P}_{a_i \sim \pi_i^k}(f \in a_i) \geq \frac{\gamma}{2F}$ for any $f \in \mathcal{F}_i$.*

*Proof.* By Algorithm 4, whenever a new action is added into $\widetilde{\mathcal{A}}_i$, it contains facility not appeared in current $\widetilde{\mathcal{A}}_i$. Then, since there are at most $|\mathcal{F}_i| \leq F$ distinct facilities in the action set $\mathcal{A}_i$, the final $\widetilde{\mathcal{A}}_i$ must satisfy $|\widetilde{\mathcal{A}}_i| \leq F$. Therefore, $\tilde{\rho}_i$ is a valid distribution over $\mathcal{A}_i$.

Since $\pi_i^k$ contains a mixture of $\tilde{\rho}_i$ with weight $\gamma$, for any $a_i \in \mathcal{A}_i$, we have $\pi_i^k(a_i) \geq \gamma\tilde{\rho}_i(a_i)$. Thus, we have

$$\mathbb{P}_{a_i \sim \pi_i^k}(f \in a_i) = \sum_{a_i \in \mathcal{A}_i} \pi_i^k(a_i)\mathbb{1}\{f \in a_i\}$$

$$\geq \gamma \sum_{a_i \in \mathcal{A}_i} \tilde{\rho}_i(a_i)\mathbb{1}\{f \in a_i\}$$

$$\geq \gamma \sum_{a_i \in \widetilde{\mathcal{A}}_i} \tilde{\rho}_i(a_i)\mathbb{1}\{f \in a_i\}$$

$$= \frac{\gamma}{2F} \sum_{a_i \in \widetilde{\mathcal{A}}_i} \mathbb{1}\{f \in a_i\} \geq \frac{\gamma}{2F}.$$

The last inequality above holds since by construction, $\widetilde{\mathcal{A}}_i$ contains all facilities contained in $\mathcal{A}_i$.

$\square$

**Lemma 15.** *If $\pi_i^k$ contains a mixture of $\tilde{\rho}_i$ given in Algorithm 4 with weight $\gamma$. Then, the IPS estimator $[\widetilde{\theta}_i^k(\pi^k)]_f$ satisfies*

$$\mathbb{E}_k\left[[\widetilde{\theta}_i^{k,t}(\pi^k)]_f\right] = [\theta_i(\pi^k)]_f, \quad |[\widetilde{\theta}_i^{k,t}(\pi^k)]_f| \leq \frac{2F}{\gamma}, \quad \text{and} \quad \mathbb{E}_k\left[[\widetilde{\theta}_i^{k,t}(\pi^k)]_f^2\right] \leq \frac{2F}{\gamma}.$$

*Proof.* For the first property, since $\mathbb{E}_k\left[r^{k,t,f} \mid \boldsymbol{a}^{k,t}\right] = r^f(n^f(a_i^{k,t}, a_{-i}^{k,t}))$ and $\boldsymbol{a}^{k,t} \sim \pi^k$, We have

$$
\begin{aligned}
&\mathbb{E}_k\left[[\widetilde{\theta}_i^{k,t}(\pi^k)]_f\right] \\
=&\mathbb{E}_{\boldsymbol{a}\sim\pi^k}\left[\frac{r^f(n^f(a_i, a_{-i}))\mathbb{1}\{f \in a_i\}}{\mathbb{P}_{a_i'\sim\pi_i^k}(f \in a_i')}\right] \\
=&\frac{1}{\mathbb{P}_{a_i'\sim\pi_i^k}(f \in a_i')} \cdot \mathbb{E}_{a_{-i}\sim\pi_{-i}^k}\left[\mathbb{E}_{a_i\sim\pi_i^k}\left[r^f(n^f(a_i, a_{-i}))\mathbb{1}\{f \in a_i\} \mid a_{-i}\right]\right] \\
=&\frac{1}{\mathbb{P}_{a_i'\sim\pi_i^k}(f \in a_i')} \cdot \mathbb{E}_{a_{-i}\sim\pi_{-i}^k}\left[\mathbb{E}_{a_i\sim\pi_i^k}\left[r^f(n^f(a_i, a_{-i})) \mid a_{-i}, f \in a_i\right]\mathbb{P}_{a_i\sim\pi_i^k}(f \in a_i \mid a_{-i})\right] \\
\overset{(i)}{=}&\frac{\mathbb{P}_{a_i\sim\pi_i^k}(f \in a_i)}{\mathbb{P}_{a_i'\sim\pi_i^k}(f \in a_i')} \cdot \mathbb{E}_{a_{-i}\sim\pi_{-i}^k}\left[r^f(n^f(a_{-i}) + 1)\right] \\
=&[\theta_i(\pi^k)]_f.
\end{aligned}
$$

The equality (i) above holds because $\mathbb{E}_{a_i\sim\pi_i^k}\left[r^f(n^f(a_i, a_{-i})) \mid a_{-i}, f \in a_i\right] = r^f(n^f(a_{-i}) + 1)$ and $f \in a_i$ does not depend on $a_{-i}$.

For the second property, since $\mathbb{P}_{a_i\sim\pi_i^k}(f \in a_i) \geq \frac{\gamma}{2F}$ by Lemma 14 and $r^{k,t,f} \in [0,1]$, we can immediately have $|[\widetilde{\theta}_i^{k,t}(\pi^k)]_f| \leq \frac{2F}{\gamma}$.

For the third property, we have

$$
\begin{aligned}
\mathbb{E}_k\left[[\widetilde{\theta}_i^{k,t}(\pi^k)]_f^2\right] =&\frac{\mathbb{E}_{\boldsymbol{a}\sim\pi^k}\left[r^f(n^f(a_i, a_{-i}))^2\mathbb{1}\{f \in a_i\}\right]}{\mathbb{P}_{a_i'\sim\pi_i^k}(f \in a_i')^2} \\
\leq&\frac{\mathbb{E}_{\boldsymbol{a}\sim\pi^k}\left[\mathbb{1}\{f \in a_i\}\right]}{\mathbb{P}_{a_i'\sim\pi_i^k}(f \in a_i')^2} \\
=&\frac{\mathbb{P}_{a_i\sim\pi_i^k}(f \in a_i)}{\mathbb{P}_{a_i'\sim\pi_i^k}(f \in a_i')^2} \\
\leq&\frac{2F}{\gamma}.
\end{aligned}
$$

$\square$

# E  Algorithms for Independent Markov Congestion Games

In this section, present missing details of our centralized algorithm for independent Markov congestion games, which is summarized in Algorithm 5. The proof of its theoretical guarantee is given in Appendix F.

## E.1  Algorithm for Semi-bandit Feedback

Under the semi-bandit feedback, the players can receive reward information from all facilities they choose. Therefore, we can similarly define

$$N_h^{k,f}(s^f, n) = \sum_{k'=1}^k \mathbb{1}\left\{(s_h^{k',f}, n^f(\boldsymbol{a}_h^{k'})) = (s^f, n)\right\},$$

---

**Algorithm 5** Nash-VI for IMCGs

---
1: **Input:** $\epsilon$, accuracy parameter for Nash equilibrium computation
2: **Initialize:** $\overline{V}_{H+1,i}^k(s) = 0$ for all $(i,k,s) \in [m] \times [K] \times \mathcal{S}$
3: **for** episode $k = 1, \ldots, K$ **do**
4:     **for** step $h = H, H-1, \ldots, 1$ **do**
5:         **for** player $i = 1, \ldots, m$ **do**
6:             $\overline{Q}_{h,i}^k(s,\boldsymbol{a}) \leftarrow \min\left\{(\hat{r}_{h,i}^k + \widehat{\mathbb{P}}_h^k \overline{V}_{h+1,i}^k + b_h^k)(s,\boldsymbol{a}), HF\right\}$ for all $(s,\boldsymbol{a}) \in \mathcal{S} \times \mathcal{A}$
7:         **for** $s \in \mathcal{S}$ **do**
8:             $\pi_h^k(\cdot \mid s) \leftarrow \epsilon\text{-}\mathrm{NASH}(\overline{Q}_{h,1}^k(s,\cdot), \cdots, \overline{Q}_{h,m}^k(s,\cdot))$
9:             **for** player $i = 1, \ldots, m$ **do**
10:                $\overline{V}_{h,i}^k(s) \leftarrow \mathbb{E}_{\boldsymbol{a} \sim \pi_h^k}[\overline{Q}_{h,i}^k(s,\boldsymbol{a})]$
11:     **for** step $h = 1, \ldots, H$ **do**
12:         Take action $\boldsymbol{a}_h^k \sim \pi_h^k(\cdot \mid s_h^k)$, observe reward $r_h^{k,f}$ and next state $s_{h+1}^k$
13:         Update reward estimator $\hat{r}_{h,i}^k$, transition estimator $\widehat{P}_h^k$ and bonus term $b_h^k$

---

$$\hat{r}_h^{k,f}(s^f, n) = \frac{\sum_{k'=1}^k r_h^{k',f} \mathbb{1}\left\{(s_h^{k',f}, n^f(\boldsymbol{a}_h^{k'})) = (s^f, n)\right\}}{N_h^{k,f}(s^f, n) \vee 1},$$

$$\widehat{P}_h^{k,f}(s'^f \mid s^f, n) = \frac{\sum_{k'=1}^k \mathbb{1}\left\{(s_{h+1}^{k',f}, s_h^{k',f}, n^f(\boldsymbol{a}_h^{k'})) = (s'^f, s^f, n)\right\}}{N_h^{k,f}(s^f, n) \vee 1}.$$

Then, the estimators for the reward function and transition kernel can be defined as

$$\hat{r}_{h,i}^k(s, \boldsymbol{a}) = \sum_{f \in a_i} \hat{r}_h^{k,f}(s^f, n^f(\boldsymbol{a})), \quad \widehat{P}_h^k(s' \mid s, \boldsymbol{a}) = \prod_{f \in \mathcal{F}} \widehat{P}_h^{k,f}(s'^f \mid s^f, n^f(\boldsymbol{a})) \qquad (7)$$

Then, with $\iota = 2\log(4(m+1)(\sum_{f \in \mathcal{F}} S^f)T/\delta)$, we define the bonus term to be $b_h^k(s, \boldsymbol{a}) = b_h^{k,\mathrm{pv}}(s, \boldsymbol{a}) + b_h^{k,\mathrm{r}}(s, \boldsymbol{a})$, which is a sum of transition bonus and reward bonus. In particular, we have

$$b_h^{k,\mathrm{pv}}(s, \boldsymbol{a}) = \sum_{f \in \mathcal{F}} \sqrt{\frac{4H^2F^2 S^f \iota}{N_h^{k,f}(s^f, n^f(\boldsymbol{a})) \vee 1}} + \sum_{f \neq f'} \sqrt{\frac{4H^2F^2 \left(S^f S^{f'}\right)^2}{N_h^{k,f}(s^f, n^f(\boldsymbol{a})) N_h^{k,f'}(s^{f'}, n^{f'}(\boldsymbol{a})) \vee 1}}, \quad (8)$$

$$b_h^{k,\mathrm{r}}(s, \boldsymbol{a}) = \sum_{f \in \mathcal{F}} \sqrt{\frac{\iota}{N_h^{k,f}(s^f, n^f(\boldsymbol{a})) \vee 1}}. \qquad (9)$$

For convenience, we define $(\widehat{\mathbb{P}}_h^k V)(s, \boldsymbol{a}) = \mathbb{E}_{s' \sim \widehat{P}_h^k(\cdot \mid s, \boldsymbol{a})}[V(s')]$ with value function $V : \mathcal{S} \mapsto \mathbb{R}$.

*Remark* 5. Unlike Algorithm 1 for congestion game, here, $\overline{Q}_{h,1}^k(s, \cdot), \ldots, \overline{Q}_{h,m}^k(s, \cdot)$ in line 6 of Algorithm 5 in general does not form a potential game. Therefore, we cannot use Algorithm 2 and $\epsilon$-NASH is not always computationally efficient.

### E.2 Algorithm for Bandit Feedback

In bandit feedback scenario, since players' observation about state transitions remains unaffected, we only need to modify the reward estimator $\hat{r}_{h,i}^k$ defined in (7) and reward bonus term $b_h^{k,\mathrm{r}}(s, \boldsymbol{a})$ defined in (9).

Similar to the congestion game with bandit feedback introduced in Section 4.2, for IMCGs, we can also write its reward function as $r_{h,i}(s, \boldsymbol{a}) = \langle A_i(s, \boldsymbol{a}), \theta_h \rangle$, where $\theta_h$ is unknown and $A_i(s, \boldsymbol{a})$ is a 0-1 vector.

In particular, define $\theta_h \in [0, 1]^d$ with $d = m \sum_{f \in \mathcal{F}} S^f$ to be the vector such that $\theta_{h,i} = r_h^f(s^f, n)$ for some $f \in \mathcal{F}$ and $(s^f, n) \in \mathcal{S}^f \times [m]$. Then, we can similarly build estimator $\hat{r}_{h,i}^k$ through ridge regression as the following.[3]

---

[3]For the same reason, we take the regularization parameter in ridge regression to be 1.

$$\text{design matrix:} \quad V_h^k = I + \sum_{k'=1}^{k-1} \sum_{i=1}^{m} A_i(s_h^{k'}, \boldsymbol{a}_h^{k'}) A_i(s_h^{k'}, \boldsymbol{a}_h^{k'})^\top, \tag{10}$$

$$\theta_h \text{ estimator:} \quad \widehat{\theta}_h^k = \left(V_h^k\right)^{-1} \sum_{k'=1}^{k-1} \sum_{i=1}^{m} A_i(s_h^{k'}, \boldsymbol{a}_h^{k'}) r_{h,i}^{k'}, \tag{11}$$

$$\text{reward estimator:} \quad \tilde{r}_{h,i}^k(s, \boldsymbol{a}) = \left\langle A_i(s, \boldsymbol{a}), \widehat{\theta}_h^k \right\rangle, \tag{12}$$

$$\text{reward bonus:} \quad \tilde{b}_h^{k,\mathrm{r}}(s, \boldsymbol{a}) = \max_{i \in [m]} \|A_i(s, \boldsymbol{a})\|_{(V_h^k)^{-1}} \sqrt{\beta_k}, \tag{13}$$

where $\sqrt{\beta_k} = \sqrt{d} + \sqrt{Fd \log\left(1 + \frac{mkF}{d}\right) + F\iota}$.

# F   Analysis for Algorithm 5

## F.1   Bellman Equations for Genera-sum Markov Games

Before analyzing Algorithm 5, we first give a brief review of the Bellman equations for general-sum Markov games. These equations are well-known among the literature Bai and Jin [2020], Liu et al. [2021], Jin et al. [2021b].

**Fixed policies.**   Given a fixed policy $\pi$, for any $(h, i, s, \boldsymbol{a}) \in [H] \times [m] \times \mathcal{S} \times \mathcal{A}$, it holds that

$$Q_{h,i}^\pi(s, \boldsymbol{a}) = (r_{h,i} + \mathbb{P}_h V_{h+1,i}^\pi)(s, \boldsymbol{a}), \quad V_{h,i}^\pi = \mathbb{E}_{\boldsymbol{a}' \sim \pi_h(\cdot|s)} \left[Q_{h,i}^\pi(s, \boldsymbol{a}')\right], \tag{14}$$

where $V_{H+1,i}^\pi(s) = 0$ for any $(i, s) \in [m] \times \mathcal{S}$.

**Best responses.**   Given a fixed policy $\pi$, define the best response value functions for player $i$ as $Q_{h,i}^{\dagger, \pi_{-i}}(s, \boldsymbol{a}) = \max_{\pi_i \in \Delta(\mathcal{A}_i)} Q_{h,i}^{\pi_i, \pi_{-i}}(s, \boldsymbol{a})$ and $V_{h,i}^{\dagger, \pi_{-i}}(s) = \max_{\pi_i \in \Delta(\mathcal{A}_i)} V_{h,i}^{\pi_i, \pi_{-i}}(s)$. Then, for any $(h, i, s, \boldsymbol{a}) \in [H] \times [m] \times \mathcal{S} \times \mathcal{A}$, it holds that

$$\begin{aligned}
Q_{h,i}^{\dagger, \pi_{-i}}(s, \boldsymbol{a}) &= (r_{h,i} + \mathbb{P}_h V_{h+1,i}^{\dagger, \pi_{-i}})(s, \boldsymbol{a}), \\
V_{h,i}^{\dagger, \pi_{-i}}(s) &= \max_{\nu \in \Delta(\mathcal{A}_i)} \mathbb{E}_{\boldsymbol{a}' \sim (\nu, \pi_{h,-i})(\cdot|s)} \left[Q_{h,i}^{\dagger, \pi_{-i}}(s, \boldsymbol{a}')\right],
\end{aligned} \tag{15}$$

where $V_{H+1,i}^{\dagger, \pi_{-i}}(s) = 0$ for any $(i, s) \in [m] \times \mathcal{S}$.

## F.2   Proof of Theorem 3

Recall that the update rule in Algorithm 5 is

$$\overline{Q}_{h,i}^k(s, \boldsymbol{a}) \leftarrow \min \left\{ (\hat{r}_{h,i}^k + \widehat{\mathbb{P}}_h^k \overline{V}_{h+1,i}^k + b_h^k)(s, \boldsymbol{a}), HF \right\}, \quad \overline{V}_{h,i}^k(s) \leftarrow \mathbb{E}_{\boldsymbol{a} \sim \pi_h^k}[\overline{Q}_{h,i}^k(s, \boldsymbol{a})].$$

Similar to the proof of Theorem 1, we define auxiliary value functions

$$\underline{Q}_{h,i}^k(s, \boldsymbol{a}) \leftarrow \max \left\{ (\hat{r}_{h,i}^k + \widehat{\mathbb{P}}_h^k \underline{V}_{h+1,i}^k - b_h^k)(s, \boldsymbol{a}), 0 \right\}, \quad \underline{V}_{h,i}^k(s) \leftarrow \mathbb{E}_{\boldsymbol{a} \sim \pi_h^k}[\underline{Q}_{h,i}^k(s, \boldsymbol{a})]. \tag{16}$$

We now begin to prove the first part of Theorem 3.

*Proof of Theorem 3.* **Step 1.** We first consider the setting of semi-bandit feedback. Assume the result in Lemma 17 holds since it is a high-probability event. Then, for any $(k, s) \in [K] \times \mathcal{S}$, it holds that

$$\max_{i \in [m]} \left(V_{1,i}^{\dagger, \pi_{-i}^k} - V_{1,i}^{\pi^k}\right)(s) \leq \max_{i \in [m]} \left(\overline{V}_{1,i}^k - \underline{V}_{1,i}^k\right)(s) + H\epsilon.$$

By the update rules in Algorithm 5, we can notice the following recursive relations

$$(\overline{Q}_{h,i}^k - \underline{Q}_{h,i}^k)(s, \boldsymbol{a}) \leq \min \left\{ \widehat{\mathbb{P}}_h^k (\overline{V}_{h+1,i}^k - \underline{V}_{h+1,i}^k)(s, \boldsymbol{a}) + 2b_h^k(s, \boldsymbol{a}), HF \right\},$$

$$(\overline{V}_{h,i}^k - \underline{V}_{h,i}^k)(s) = \mathbb{E}_{\boldsymbol{a}' \sim \pi_h^k(\cdot|s)}\left[(\overline{Q}_{h,i}^k - \underline{Q}_{h,i}^k)(s, \boldsymbol{a}')\right].$$

Thus, we define $\widetilde{V}_{H+1}^k(s) = 0$ for any $s \in \mathcal{S}$ and $\widetilde{Q}_h^k, \widetilde{V}_h^k$ recursively as

$$\widetilde{Q}_h^k(s, \boldsymbol{a}) = \min\left\{(\widehat{\mathbb{P}}_h^k \widetilde{V}_{h+1}^k)(s, \boldsymbol{a}) + 2b_h^k(s, \boldsymbol{a}), HF\right\}, \quad \widetilde{V}_h^k(s) = \mathbb{E}_{\boldsymbol{a}' \sim \pi_h^k(\cdot|s)}\left[\widetilde{Q}_h^k(s, \boldsymbol{a}')\right]. \quad (17)$$

Obviously, we have $\max_{i \in [m]}(\overline{V}_{h,i}^k - \underline{V}_{h,i}^k)(s) \leq \widetilde{V}_{H+1}^k$. Then, by inductively assuming the same relation holds for $h + 1$, we can have

$$\max_{i \in [m]}(\overline{Q}_{h,i}^k - \underline{Q}_{h,i}^k)(s, \boldsymbol{a}) = \min\left\{\max_{i \in [m]}\widehat{\mathbb{P}}_h^k(\overline{V}_{h+1,i}^k - \underline{V}_{h+1,i}^k)(s, \boldsymbol{a}) + 2b_h^k(s, \boldsymbol{a}), HF\right\}$$

$$\leq \min\left\{(\widehat{\mathbb{P}}_h^k \widetilde{V}_{h+1}^k)(s, \boldsymbol{a}) + 2b_h^k(s, \boldsymbol{a}), HF\right\}$$

$$= \widetilde{Q}_h^k(s, \boldsymbol{a}),$$

$$\max_{i \in [m]}(\overline{V}_{h,i}^k - \underline{V}_{h,i}^k)(s) \leq \mathbb{E}_{\boldsymbol{a}' \sim \pi_h^k(\cdot|s)}\left[\max_{i \in [m]}(\overline{Q}_{h,i}^k - \underline{Q}_{h,i}^k)(s, \boldsymbol{a}')\right]$$

$$\leq \mathbb{E}_{\boldsymbol{a}' \sim \pi_h^k(\cdot|s)}\left[\widetilde{Q}_h^k(s, \boldsymbol{a}')\right]$$

$$= \widetilde{V}_h^k(s).$$

Therefore, by induction, for any $h \in [H]$, we have

$$\max_{i \in [m]}(\overline{Q}_{h,i}^k - \underline{Q}_{h,i}^k)(s, \boldsymbol{a}) \leq \widetilde{Q}_h^k(s, \boldsymbol{a}), \quad \max_{i \in [m]}(\overline{V}_{h,i}^k - \underline{V}_{h,i}^k)(s) \leq \widetilde{V}_h^k(s).$$

As a result, we have

$$\text{Nash-Regret}(K) = \sum_{k=1}^K \max_{i \in [m]}\left(V_{1,i}^{\dagger, \pi_{-i}^k} - V_{1,i}^{\pi^k}\right)(s) \leq \sum_{k=1}^K \widetilde{V}_1^k(s_1) + HK\epsilon.$$

**Step 2, Semi-bandit Feedback.** We define the martingale difference sequences

$$\mathcal{M}_h^k(\widetilde{Q}) = \mathbb{E}_{\boldsymbol{a}' \sim \pi_h^k(\cdot|s_h^k)}\left[\widetilde{Q}_h^k(s_h^k, \boldsymbol{a}')\right] - \widetilde{Q}_h^k(s_h^k, \boldsymbol{a}_h^k),$$

$$\mathcal{M}_h^k(\widetilde{V}) = (\mathbb{P}_h \widetilde{V}_{h+1}^k)(s_h^k, \boldsymbol{a}_h^k) - \widetilde{V}_{h+1}^k(s_{h+1}^k).$$

It is not hard to check that $\mathcal{M}_h^k(\widetilde{Q})$ and $\mathcal{M}_h^k(\widetilde{V})$ are both indeed martingale difference sequences with respect to the history till episode $k$ and time step $h$.

With these definitions, we can now decompose the regret bound as

$$\widetilde{V}_h^k(s_h^k) = \mathbb{E}_{\boldsymbol{a}' \sim \pi_h^k(\cdot|s_h^k)}\left[\widetilde{Q}_h^k(s_h^k, \boldsymbol{a}')\right] \qquad\qquad \text{(By (17))}$$

$$= \mathcal{M}_h^k(\widetilde{Q}) + \widetilde{Q}_h^k(s_h^k, \boldsymbol{a}_h^k)$$

$$\leq \mathcal{M}_h^k(\widetilde{Q}) + 2b_h^k(s_h^k, \boldsymbol{a}_h^k) + (\widehat{\mathbb{P}}_h^k \widetilde{V}_{h+1}^k)(s_h^k, \boldsymbol{a}_h^k) \qquad\qquad \text{(By (17))}$$

$$\overset{(i)}{\leq} \mathcal{M}_h^k(\widetilde{Q}) + 3b_h^k(s_h^k, \boldsymbol{a}_h^k) + (\mathbb{P}_h \widetilde{V}_{h+1}^k)(s_h^k, \boldsymbol{a}_h^k)$$

$$= \mathcal{M}_h^k(\widetilde{Q}) + \mathcal{M}_h^k(\widetilde{V}) + 3b_h^k(s_h^k, \boldsymbol{a}_h^k) + \widetilde{V}_{h+1}^k(s_{h+1}^k)$$

The above inequality (i) holds by applying Lemma 17 and the fact $\widetilde{V}_h^k(s) \leq HF$, which comes from the definition in (17). Then, by unrolling this relation from $h = 1$ to $h = H$ and noticing $\widetilde{V}_{H+1}^k = \mathbf{0}$, we can have

$$\text{Nash-Regret}(K) \leq \sum_{k=1}^K \widetilde{V}_1^k(s_1) + HK\epsilon$$

$$\leq \sum_{k=1}^K \sum_{h=1}^H \left(\mathcal{M}_h^k(\widetilde{Q}) + \mathcal{M}_h^k(\widetilde{V}) + 3b_h^k(s_h^k, \boldsymbol{a}_h^k)\right) + HK\epsilon \qquad\qquad (18)$$

$$\leq \widetilde{\mathcal{O}}\left(HF\sqrt{T}\right) + 3\sum_{k=1}^{K}\sum_{h=1}^{H} b_h^k(s_h^k, \boldsymbol{a}_h^k) \qquad \text{(By Azuma-Hoeffding inequality and taking } \epsilon = 1/T.)$$

$$\leq \widetilde{\mathcal{O}}\left(HF\sqrt{T}\right) + 6HF\sum_{f\in\mathcal{F}}\sum_{k=1}^{K}\sum_{h=1}^{H}\left(\sqrt{\frac{S^f\iota}{N_h^{k,f}(s_h^{k,f}, n^f(\boldsymbol{a}_h^k))\vee 1}} + \sqrt{\frac{\iota}{N_h^{k,f}(s_h^{k,f}, n^f(\boldsymbol{a}_h^k))\vee 1}}\right)$$

$$+ 6HF\sum_{f\neq f'} S^f S^{f'}\sum_{k=1}^{K}\sum_{h=1}^{H}\sqrt{\frac{\iota^2}{\left(N_h^{k,f}(s_h^{k,f}, n^f(\boldsymbol{a}_h^k))N_h^{k,f'}(s_h^{k,f'}, n^{f'}(\boldsymbol{a}_h^{k,f'}))\right)\vee 1}}$$

$$\leq \widetilde{\mathcal{O}}\left(HF\sqrt{T}\right) + \widetilde{\mathcal{O}}\left(\sum_{f\in\mathcal{F}} HFS^f\sqrt{mHT}\right) + \widetilde{\mathcal{O}}\left(m^2H^2F\sum_{f\neq f'}\left(S^f S^{f'}\right)^2\right)$$

$$\text{(By Lemma 20 and 21)}$$

$$\leq \widetilde{\mathcal{O}}\left(\sum_{f\in\mathcal{F}} FS^f\sqrt{mH^3T}\right) + \widetilde{\mathcal{O}}\left(m^2H^2F\sum_{f\neq f'}\left(S^f S^{f'}\right)^2\right).$$

**Step 3, Bandit Feedback.** In the setting of bandit feedback, we only modify the reward estimator $\tilde{r}_{h,i}^k$ and its corresponding bonus term $\tilde{b}_h^{k,\text{r}}$. Thus, by going through the proof of Lemma 17, we can notice that to have the same result for bandit feedback, it suffice to use Lemma 18 to show that the reward estimation error is bounded by the reward bonus term.

Then, by the inequality (18), we can notice that to achieve the final Nash-regret bound, we only need to bound the summation $\sum_{k=1}^{K}\sum_{h=1}^{H}\tilde{b}_h^{k,\text{r}}(s_h^k, \boldsymbol{a}_h^k)$, which is

$$\sum_{k=1}^{K}\sum_{h=1}^{H}\tilde{b}_h^{k,\text{r}}(s_h^k, \boldsymbol{a}_h^k) \leq \sqrt{\beta_K}\sum_{k=1}^{K}\sum_{h=1}^{H}\max_{i\in[m]}\left\|A_i(s_h^k, \boldsymbol{a}_h^k)\right\|_{(V_h^k)^{-1}} \qquad \text{(By definition of } \tilde{b}_h^{k,\text{r}} \text{ in (13).)}$$

$$\leq \left(\sqrt{d} + \sqrt{Fd\log\left(1 + \frac{mKF}{d}\right) + F\iota}\right)\widetilde{\mathcal{O}}\left(H\sqrt{dFK}\right)$$

$$\text{(By definition of } \beta_k \text{ and Lemma 19.)}$$

$$\leq \widetilde{\mathcal{O}}\left(d\sqrt{HF^2T}\right)$$

$$= \widetilde{\mathcal{O}}\left(\sum_{f\in\mathcal{F}} mS^f\sqrt{HF^2T}\right). \qquad \text{(Since } d = m\sum_{f\in\mathcal{F}} S^f.)$$

Therefore, by (18), with $\epsilon = 1/T$, under bandit feedback, we have

$$\text{Nash-Regret}(K)$$

$$\leq \sum_{k=1}^{K}\sum_{h=1}^{H}\left(\mathcal{M}_h^k(\widetilde{Q}) + \mathcal{M}_h^k(\widetilde{V}) + 3b_h^k(s_h^k, \boldsymbol{a}_h^k)\right)$$

$$\leq \widetilde{\mathcal{O}}\left(\sum_{f\in\mathcal{F}} FS^f\sqrt{mH^3T}\right) + \widetilde{\mathcal{O}}\left(m^2H^2F\sum_{f\neq f'}\left(S^f S^{f'}\right)^2\right) + \sum_{k=1}^{K}\sum_{h=1}^{H}\tilde{b}_h^{k,\text{r}}(s_h^k, \boldsymbol{a}_h^k)$$

$$\leq \widetilde{\mathcal{O}}\left(\sum_{f\in\mathcal{F}}\left(\sqrt{mH^3}F + m\sqrt{HF^2}\right)S^f\sqrt{T}\right) + \widetilde{\mathcal{O}}\left(m^2H^2F\sum_{f\neq f'}\left(S^f S^{f'}\right)^2\right).$$

$$\square$$

### F.3 Lemmas for Semi-bandit Feedback

The following two lemmas shows that our value function estimations are indeed optimistic.

**Lemma 16.** *With probability at least $1-\delta$, simultaneously for arbitrary value function $V \in [0, HF]^{\mathcal{S}}$ and any tuple $(k, h, s, \boldsymbol{a})$, it holds that $|(\widehat{\mathbb{P}}_h^k - \mathbb{P}_h)V(s, \boldsymbol{a})| \leq b_h^{k,\text{pv}}(s, \boldsymbol{a})$, where $b_h^{k,\text{pv}}(s, \boldsymbol{a})$ is defined in (8).*

*Proof.* We define $\mathbb{P}_h^f$ to be the operator such that for some value function $V^f : \mathcal{S}^f \mapsto \mathbb{R}$, we have $(\mathbb{P}_h^f V^f)(s, \boldsymbol{a}) = \mathbb{E}_{s'^f \sim P_h^f(\cdot \mid s^f, n^f(\boldsymbol{a}))} \left[ V^f(s'^f) \right]$. We also define $\widehat{\mathbb{P}}_h^{k,f}$ similarly. Then, by definition of our transition kernel, for operators $\mathbb{P}_h$ and $\widehat{\mathbb{P}}_h^k$, it holds that

$$\mathbb{P}_h = \prod_{f \in \mathcal{F}} \mathbb{P}_h^f \quad \text{and} \quad \widehat{\mathbb{P}}_h^k = \prod_{f \in \mathcal{F}} \widehat{\mathbb{P}}_h^{k,f}.$$

Therefore, by Lemma E.1 in Chen et al. [2020], since $\|V\|_\infty \leq HF$, we have

$$
\begin{aligned}
|(\widehat{\mathbb{P}}_h^k - \mathbb{P}_h)V(s, \boldsymbol{a})| \leq \sum_{f \in \mathcal{F}} &\left| (\widehat{\mathbb{P}}_h^{k,f} - \mathbb{P}_h^f) \left( \prod_{f' \neq f} \mathbb{P}_h^{f'} \right) V(s, \boldsymbol{a}) \right| \\
&+ 2HF \sum_{f \neq f'} \mathrm{errp}_h^{k,f}(s, \boldsymbol{a}) \cdot \mathrm{errp}_h^{k,f'}(s, \boldsymbol{a}),
\end{aligned}
\tag{19}
$$

where $\mathrm{errp}_h^{k,f}(s, \boldsymbol{a}) = \|\widehat{P}_h^{k,f}(\cdot \mid s^f, n^f(\boldsymbol{a})) - P_h^f(\cdot \mid s^f, n^f(\boldsymbol{a}))\|_1$.

Now, notice that $\left( \prod_{f' \neq f} \mathbb{P}_h^{f'} \right) V(s, \boldsymbol{a})$ can be seen as some value function from $\mathcal{S}^f$ to $[0, HF]$. Therefore, by Lemma 12 in Bai and Jin [2020], with probability at least $1 - \frac{\delta}{2}$, simultaneously for any $V$ and $(k, h, s, \boldsymbol{a})$, it holds that

$$\left| (\widehat{\mathbb{P}}_h^{k,f} - \mathbb{P}_h^f) \left( \prod_{f' \neq f} \mathbb{P}_h^{f'} \right) V(s, \boldsymbol{a}) \right| \leq 2HF \sqrt{\frac{S^f \iota}{N_h^{k,f}(s^f, n^f(\boldsymbol{a})) \vee 1}},$$

where $\iota = 2\log(4(m+1)(\sum_{f \in \mathcal{F}} S^f)T/\delta)$. Meanwhile, by standard Hoeffding's inequality and union bound, with probability at least $1 - \frac{\delta}{2}$, simultaneously for any $(k, h, s, \boldsymbol{a})$, it holds that

$$\mathrm{errp}_h^{k,f} \leq S^f \sqrt{\frac{\iota}{N_h^{k,f}(s^f, n^f(\boldsymbol{a})) \vee 1}}.$$

Finally, by plugging above two concentration inequalities back into (19), we can have

$$|(\widehat{\mathbb{P}}_h^k - \mathbb{P}_h)V(s, \boldsymbol{a})| \leq b_h^{k,\mathrm{pv}}(s, \boldsymbol{a}).$$

$\square$

**Lemma 17.** *With probability at least $1 - \delta$, for any $(k, h, i, s, \boldsymbol{a}) \in [K] \times [H] \times [m] \times \mathcal{S} \times \mathcal{A}$, it holds that*

$$\overline{Q}_{h,i}^k(s, \boldsymbol{a}) \geq Q_{h,i}^{\dagger, \pi_{-i}^k}(s, \boldsymbol{a}) - (H - h)\epsilon, \quad \underline{Q}_{h,i}^k(s, \boldsymbol{a}) \leq Q_{h,i}^{\pi^k}(s, \boldsymbol{a}), \tag{20}$$

$$\overline{V}_{h,i}^k(s) \geq V_{h,i}^{\dagger, \pi_{-i}^k}(s) - (H - h + 1)\epsilon, \quad \underline{V}_{h,i}^k(s) \leq V_{h,i}^{\pi^k}(s.), \tag{21}$$

*where $\underline{Q}_{h,k}^k$ and $\underline{V}_{h,i}^k$ are defined in (16).*

*Proof.* The proof is adapted from Liu et al. [2021] and goes by induction from $h = H + 1$ to $h = 1$. We can see that inequalities (21) obviously hold when $h = H + 1$ since by definition we have $\overline{V}_{H+1,i}^k(s) = \underline{V}_{H+1,i}^k(s) = 0$ for any $(k, i, s)$. Now, suppose inequalities (21) hold for $h+1$. Then, if we have $\overline{Q}_{h,i}^k(s, \boldsymbol{a}) = HF$, it holds trivially that $\overline{Q}_{h,i}^k(s, \boldsymbol{a}) \geq Q_{h,i}^{\dagger, \pi_{-i}^k}(s, \boldsymbol{a})$. Otherwise, by Bellman equations (15) and update rule in Algorithm 5, we have

$$
\begin{aligned}
&\overline{Q}_{h,i}^k(s, \boldsymbol{a}) - Q_{h,i}^{\dagger, \pi_{-i}^k}(s, \boldsymbol{a}) \\
=& (\hat{r}_{h,i}^k - r_{h,i})(s, \boldsymbol{a}) + (\widehat{\mathbb{P}}_h^k \overline{V}_{h+1,i}^k)(s, \boldsymbol{a}) - (\mathbb{P}_h V_{h+1,i}^{\dagger, \pi_{-i}^k})(s, \boldsymbol{a}) + b_h^k(s, \boldsymbol{a}) \\
=& \underbrace{(\hat{r}_{h,i}^k - r_{h,i})(s, \boldsymbol{a})}_{(A)} + \underbrace{\widehat{\mathbb{P}}_h^k (\overline{V}_{h+1,i}^k - V_{h+1,i}^{\dagger, \pi_{-i}^k})(s, \boldsymbol{a})}_{(B)} + \underbrace{((\widehat{\mathbb{P}}_h^k - \mathbb{P}_h)V_{h+1,i}^{\dagger, \pi_{-i}^k})(s, \boldsymbol{a})}_{(C)} + b_h^k(s, \boldsymbol{a}).
\end{aligned}
$$

Now, recall that $b_h^k(s, \boldsymbol{a}) = b_h^{k,\mathrm{pv}}(s, \boldsymbol{a}) + b_h^{k,\mathrm{r}}(s, \boldsymbol{a})$. By reward definition in congestion game, we have

$$(\hat{r}_{h,i}^k - r_{h,i})(s, \boldsymbol{a}) = \sum_{f \in a_i} (\hat{r}_{h,i}^{k,f}(s^f, n^f(\boldsymbol{a})) - r_{h,i}^f(s^f, n^f(\boldsymbol{a}))).$$

Thus, by using standard Hoefding's inequality and union bound, we can immediately have $|(A)| \leq b_h^{k,\mathrm{r}}(s, \boldsymbol{a})$. Then, since $V_{h,i}^{\dagger,\pi_{-i}^k} \in [0, HF]^{\mathcal{S}}$, by Lemma 16, we have $|(C)| \leq b_h^{k,\mathrm{pv}}(s, \boldsymbol{a})$. That is, we have $(A) + (C) + b_h^k(s, \boldsymbol{a}) \geq 0$.

Then, by inductive hypothesis, we know that $\overline{V}_{h+1,i}^k \geq V_{h+1,i}^{\dagger,\pi_{-i}^k} - (H-h)\epsilon$, which implies $(B) \geq 0$. Therefore, we have $\overline{Q}_{h,i}^k(s, \boldsymbol{a}) - Q_{h,i}^{\dagger,\pi_{-i}^k}(s, \boldsymbol{a}) \geq -(H-h)\epsilon$.

For $\overline{V}_{h,i}^k$ and $V_{h,i}^{\dagger,\pi_{-i}^k}$, we notice that in Algorithm 5, $\pi^k$ is computed as the $\epsilon$-approximate Nash equilibrium of $(\overline{Q}_{h,1}^k, \ldots, \overline{Q}_{h,m}^k)$. Therefore, it holds that

$$\overline{V}_{h,i}^k(s) = \mathbb{E}_{\boldsymbol{a} \sim \pi_h^k(\cdot|s)} \left[ \overline{Q}_{h,i}^k(s, \boldsymbol{a}) \right] \geq \max_{\nu \in \Delta(\mathcal{A}_i)} \mathbb{E}_{\boldsymbol{a}' \sim (\nu, \pi_{h,-i}^k)(\cdot|s)} \left[ \overline{Q}_{h,i}^k(s, \boldsymbol{a}') \right] - \epsilon.$$

By Bellman equations (15), we also have

$$V_{h,i}^{\dagger,\pi_{-i}^k}(s) = \max_{\nu \in \Delta(\mathcal{A}_i)} \mathbb{E}_{\boldsymbol{a}' \sim (\nu, \pi_{h,-i}^k)(\cdot|s)} \left[ Q_{h,i}^{\dagger,\pi_{-i}^k}(s, \boldsymbol{a}') \right].$$

Since $\overline{Q}_{h,i}^k(s, \boldsymbol{a}) - Q_{h,i}^{\dagger,\pi_{-i}^k}(s, \boldsymbol{a}) \geq -(H-h)\epsilon$, we immediately have $\overline{V}_{h,i}^k(s) - V_{h,i}^{\dagger,\pi_{-i}^k}(s) \geq -(H-h+1)\epsilon$. Thus, by induction, we have that $\overline{Q}_{h,i}^k(s, \boldsymbol{a}) \geq Q_{h,i}^{\dagger,\pi_{-i}^k}(s, \boldsymbol{a}) - (H-h)\epsilon$ and $\overline{V}_{h,i}^k(s) \geq V_{h,i}^{\dagger,\pi_{-i}^k}(s) - (H-h+1)\epsilon$ for all $h \in [H]$.

The inequalities for $\underline{V}_{h,i}^k$ and $\underline{Q}_{h,i}^k$ can be proved similarly. $\qquad\square$

### F.4 Additional Lemmas for Bandit Feedback

The following lemma shows that the reward estimation error can be bounded by the reward bonus term.

**Lemma 18.** *With probability at least $1 - \delta$, simultaneously for all $(i, k, h, s, \boldsymbol{a})$, it holds that $|(\tilde{r}_{h,i}^k - r_{h,i})(s, \boldsymbol{a})| \leq \tilde{b}_h^{k,\mathrm{r}}(s, \boldsymbol{a})$, where $\tilde{r}_{h,i}^k$ and $\tilde{b}_h^{k,\mathrm{r}}$ are defined in (12) and (13).*

*Proof.* The proof is extremely similar to Lemma 3. By construction, we have

$$
\begin{aligned}
|(\tilde{r}_{h,i}^k - r_{h,i})(s, \boldsymbol{a})| &= \left| \left\langle A_i(s, \boldsymbol{a}), \widehat{\theta}_h - \theta_h \right\rangle \right| \\
&\leq \|A_i(s, \boldsymbol{a})\|_{(V_h^k)^{-1}} \left\| \widehat{\theta}_h - \theta_h \right\|_{V_h^k} \\
&\leq \|A_i(s, \boldsymbol{a})\|_{(V_h^k)^{-1}} \left( \|\theta_h\|_2 + \sqrt{F \log \left( \det(V_h^k) \right) + F\iota} \right).
\end{aligned}
$$
(By Theorem 20.5 in Lattimore and Szepesvári [2020].)

Since each element in $\theta_h$ is bounded in $[0, 1]$ by construction, we have $\|\theta_h\|_2 \leq \sqrt{d}$.

Then, by Lemma 4, we have $\det\left(V_h^k\right) \leq \left(1 + \frac{mkF}{d}\right)^d$ since by construction $\|A_i(s, \boldsymbol{a})\|_2^2 \leq F$.

Finally, to make this bound valid for all player $i \in [m]$, we only need to take maximization over $i \in [m]$. Therefore, with probability at least $1 - \delta$, we have

$$|(\tilde{r}_{h,i}^k - r_{h,i})(s, \boldsymbol{a})| \leq \max_{i \in [m]} \|A_i(s, \boldsymbol{a})\|_{(V_h^k)^{-1}} \sqrt{\beta_k} = \tilde{b}_h^{k,\mathrm{r}}(s, \boldsymbol{a}),$$

where $\sqrt{\beta_k} = \sqrt{d} + \sqrt{Fd \log \left( 1 + \frac{mkF}{d} \right) + F\iota}$. $\qquad\square$

The follow lemma bound the sum of reward bonus under bandit feedback.

**Lemma 19.** *For any $h \in [H]$, it holds that*

$$\sum_{k=1}^{K} \max_{i \in [m]} \left\| A_i(s_h^k, \boldsymbol{a}_h^k) \right\|_{(V_h^k)^{-1}} \leq \tilde{\mathcal{O}}\left(\sqrt{dFK}\right),$$

*where $d = m \sum_{f \in \mathcal{F}} S^f$.*

*Proof.* First, since $V_h^k = I + \sum_{k'=1}^{k-1} \sum_{i=1}^{m} A_i(s_h^{k'}, \boldsymbol{a}_h^{k'}) A_i(s_h^{k'}, \boldsymbol{a}_h^{k'})^\top$, we have $V_h^k \succeq I$ and thus $\left(V_h^k\right)^{-1} \preceq I$. Therefore, we have

$$\left\| A_i(s_h^k, \boldsymbol{a}_h^k) \right\|_{(V_h^k)^{-1}} \leq \left\| A_i(s_h^k, \boldsymbol{a}_h^k) \right\|_I = \left\| A_i(s_h^k, \boldsymbol{a}_h^k) \right\|_2 \leq \sqrt{F}.$$

For simplicity, let $A_{h,i}^k = A_i(s_h^k, \boldsymbol{a}_h^k)$. Then, as a result, we have

$$
\begin{aligned}
\sum_{k=1}^{K} \max_{i \in [m]} \left\| A_{h,i}^k \right\|_{(V_h^k)^{-1}} &= \sum_{k=1}^{K} \min\left\{ \max_{i \in [m]} \left\| A_{h,i}^k \right\|_{(V_h^k)^{-1}}, \sqrt{F} \right\} \\
&\leq \sqrt{ K \sum_{k=1}^{K} \min\left\{ \max_{i \in [m]} \left\| A_{h,i}^k \right\|_{(V_h^k)^{-1}}^2, F \right\} } \\
&\leq \sqrt{ FK \sum_{k=1}^{K} \min\left\{ \max_{i \in [m]} \left\| A_{h,i}^k \right\|_{(V_h^k)^{-1}}^2, 1 \right\} } \\
&\leq \sqrt{ 2FKd \log\left(1 + \frac{mKF}{d}\right) } \qquad \text{(By Lemma 4.)} \\
&= \tilde{\mathcal{O}}\left(\sqrt{dFK}\right).
\end{aligned}
$$

$\square$

### F.5 Technical Lemmas

**Lemma 20.** *For any $f \in \mathcal{F}$, it holds that*

$$\sum_{k=1}^{K} \sum_{h=1}^{H} \sqrt{ \frac{1}{N_h^{k,f}(s_h^{k,f}, n^f(\boldsymbol{a}_h^k)) \vee 1} } \leq \tilde{\mathcal{O}}\left(\sqrt{mHS^f T}\right).$$

*Proof.* Here, we have

$$
\begin{aligned}
\sum_{k=1}^{K} \sum_{h=1}^{H} \sqrt{ \frac{1}{N_h^{k,f}(s_h^{k,f}, n^f(\boldsymbol{a}_h^k)) \vee 1} } &= \sum_{h=1}^{H} \sum_{s^f \in \mathcal{S}^f} \sum_{n=0}^{m} \sum_{\ell=1}^{N_h^{K,f}(s^f, n)} \sqrt{\frac{1}{\ell}} \\
&\leq 2 \sum_{h=1}^{H} \sum_{s^f \in \mathcal{S}^f} \sum_{n=0}^{m} \sqrt{N_h^{K,f}(s^f, n)} \qquad \text{(By standard technique)} \\
&\leq 2 \sqrt{ (m+1) H S^f \sum_{h=1}^{H} \sum_{s^f \in \mathcal{S}^f} \sum_{n=0}^{m} N_h^{K,f}(s^f, n) } \\
&= \tilde{\mathcal{O}}\left(\sqrt{mHS^f T}\right).
\end{aligned}
$$

The last line above holds because $\sum_{h=1}^{H} \sum_{s^f \in \mathcal{S}^f} \sum_{n=0}^{m} N_h^{K,f}(s^f, n) = T$. This is based on a pigeon-hole principle argument. In particular, whenever the players take one more action, for any $f \in \mathcal{F}$, the count for some tuple $(h, s^f, n)$ will increase exactly by 1. $\square$

**Lemma 21** (Chen et al. [2020]). *For any $f, f' \in \mathcal{F}$ and $f \neq f'$, it holds that*

$$\sum_{k=1}^{K} \sum_{h=1}^{H} \sqrt{\frac{1}{\left(N_h^{k,f}(s_h^{k,f}, n^f(\boldsymbol{a}_h^k)) N_h^{k,f'}(s_h^{k,f'}, n^{f'}(\boldsymbol{a}_h^{k,f'}))\right) \vee 1}} \leq \widetilde{\mathcal{O}}\left(m^2 H S^f S^{f'}\right).$$

*Proof.* We define the joint empirical counter

$$N_h^{k,f,f'}(s^f, s^{f'}, n, n') = \sum_{k'=1}^{k} \mathbb{1}\left\{(s_h^{k',f}, s_h^{k',f'}, n^f(\boldsymbol{a}_h^{k'}), n^{f'}(\boldsymbol{a}_n^{k'})) = (s^f, s^{f'}, n, n')\right\}.$$

Obviously, we have $N_h^{f,f'}(s^f, s^{f'}, n, n') \leq \min\left\{N_h^{k,f}(s^f, n), N_h^{k,f'}(s^{f'}, n')\right\}$, which implies

$$N_h^{k,f,f'}(s, s^{f'}, n, n') \leq \sqrt{N_h^{k,f}(s^f, n) N_h^{k,f'}(s^{f'}, n')}.$$

Therefore, we have

$$\sum_{k=1}^{K} \sum_{h=1}^{H} \sqrt{\frac{1}{\left(N_h^{k,f}(s_h^{k,f}, n^f(\boldsymbol{a}_h^k)) N_h^{k,f'}(s_h^{k,f'}, n^{f'}(\boldsymbol{a}_h^{k,f'}))\right) \vee 1}}$$

$$\leq \sum_{k=1}^{K} \sum_{h=1}^{H} \frac{1}{N_h^{k,f,f'}(s_h^{k,f}, s_h^{k,f'}, n^f(\boldsymbol{a}_h^k), n^{f'}(\boldsymbol{a}_h^k)) \vee 1}$$

$$= \sum_{h=1}^{H} \sum_{s^f \in \mathcal{S}^f} \sum_{s^{f'} \in \mathcal{S}^{f'}} \sum_{n=0}^{m} \sum_{n'=0}^{m} \sum_{\ell=1}^{N_h^{K,f,f'}(s^f, s^{f'}, n, n')} \frac{1}{\ell}$$

$$= \widetilde{\mathcal{O}}\left(m^2 H S^f S^{f'}\right).$$

$\square$