# OpenReview forum: "Learning in Congestion Games with Bandit Feedback"
_NeurIPS.cc/2022/Conference — NeurIPS 2022 Accept_

### Official Review · Reviewer_KcbX · 2022-06-24

**Rating:** 6
**Confidence:** 3
**Soundness:** 3 good
**Presentation:** 2 fair
**Contribution:** 3 good

**Summary:**

This work studies the problem of learning in repeated congestion games and proposes the first efficient algorithms with good regret guarantees for the following cases
- centralized case with both bandit and semi-bandit feedback
- decentralized case with both bandit and semi-bandit
- centralized case for Markov congestion games with both bandit and semi-bandit feedback

**Questions:**

My main questions regard the claims of good "sample complexity" as explained in the weaknesses section.

----- Minor comments -----

1) In definition 3, I think it would be good to precise the dependency on pi of the Nash regret, as it depends on it both in the cumulated reward (second term), but also in the baseline reward (first term).

2) How can we efficiently compute the solution of eq (5) ? Also, there seems to be a typo line 283 as I believe the = sign should be an inequality?

3) In Theorems 1 and 2, what are the dependencies of the regret in delta ?

**Limitations:**

Not concerned

**Strengths And Weaknesses:**

---- Strengths ----

This work provides first good efficient algorithms for several version of online congestion games.
The used techniques are nice and natural. The claimed results seem sound.

---- Weaknesses ----

1) One of the main novelty of this paper comes from what is called "sample complexity" by the authors, but I am not even sure what they do refer by this word. It seems to me that they meant "computational complexity", but I might be wrong. In any case, there seems to be no further discussion on the "sample complexity" results after the introduction. It is a major point of this work as shown in the Table page 2, and I think the authors should explain in details (at least for one or two algorithms) why these complexity bounds do hold.

2) If my interpretation is correct and the authors did refer to computational complexity, it then seems that simulations (on a toy model) would be required to illustrate the fact that these methods can be efficiently implemented. Otherwise, this suggests that the claimed efficient algorithms are actually not that efficient.

---

> ### Author Response · Authors · 2022-08-01
> **Response**
>
> Thank you very much for your careful review and constructive suggestions! Please find our response to your questions and concerns below:
> - **Definition of sample complexity:** Here, sample complexity refers to the number of samples needed for the algorithm to obtain an $\epsilon$-good policy. In particular, the algorithm will receive reward feedback (bandit or semi-bandit) at each time step, which is referred to as a "sample". Let $T$ be the total number of time steps. Then, sample complexity means some lower bound for $T$ such that if $T\geq\text{sample complexity}$, then with high probability, the algorithm can find some policy $\pi$ such that $\max_{i\in[m]}\left(V_i^{\dagger, \pi_{-i}}-V_i^\pi\right)\leq\epsilon$. Since this lower bound for $T$ can be computed by solving $\frac{1}{T}\cdot\text{Nash regret}\leq\epsilon$, it will be sufficient to study the Nash regret. More discussion on this can be found in Section 3.1 of Jin et al. (2018). This complexity is not the same as the computational complexity and we will clarify this in the final version.
> - **Definition of Nash regret:** For policy $\pi^k$ used in episode $k$, $V_i^{\dagger,\pi_{-i}^k}$ is the value function for player $i$ such that all players $j\neq i$ use policy $\pi^k_j$ and player $i$ use the best response policy. As a result, $V_i^{\dagger,\pi_{-i}^k}-V_i^{\pi^k}$ quantifies how $\pi^k$ differs from Nash equilibrium for player $i$. Then the Nash regret at episode $k$ is $\max_{i\in[m]}\left(V_i^{\dagger,\pi_{-i}^k}-V_i^{\pi^k}\right)$ and the total Nash regret is the sum over all episodes. We will make this definition more clear in the final version.
> - **Solve equation (5):** This G-optimal design itself can be solved by another Frank-Wolfe update (See note 3 on page 269 in Lattimore et al (2020)). Since the size of action space $\mathcal{A}_i$ is exponential by definition, this computation cannot be fully efficient. However, the good thing is that $\rho_i$ is unchanged throughout the algorithm, meaning that we only need to solve it once. Meanwhile, the equality sign in line 283 is not a typo and it holds because the dimension of feature map $\phi_i$ is $F_i$. This result of G-optimal design can be found in Theorem 21.1 in Lattimore et al (2020).
> - **Dependence of $\delta$:** In Theorem 1 and 2, the regret bounds' dependence of $\delta$ is in an order of $\log(1/\delta)$, which is omitted by the $\widetilde{O}(\cdot)$ notation. We will add a reminder of this point in the final version.
>
> ```
> [Jin et al. (2018)] Chi Jin, Zeyuan Allen-Zhu, Sebastien Bubeck, and Michael I Jordan. Is Q-learning provably efficient? In Advances in Neural Information Processing Systems, pages 4863–4873, 2018.
> [Lattimore et al. (2020)] Tor Lattimore and Csaba Szepesvári. Bandit algorithms. Cambridge University Press, 2020.
> ```

---

> > ### Comment · Reviewer_KcbX · 2022-08-03
> > **Thanks for your answer**
> >
> > I want to thank the authors for their clarification on the sample complexity. Now we are clear on this point, I do not understand why the sample complexity bounds given in Table 1 are never proved in the paper. As it is a major strength of this paper, I guess such results should be theoretically proven (at least in the Appendix) and given in the main theorems.

---

> > > ### Author Response · Authors · 2022-08-04
> > > **Further explanation**
> > >
> > > Thanks for your suggestion! The sample complexity results follow directly from the standard online-to-batch conversion, e.g., as shown in Section 3.1 of [Jin et al. (2018)]. We will add the full details in the final version.
> > >
> > > ```
> > > [Jin et al. (2018)] Chi Jin, Zeyuan Allen-Zhu, Sebastien Bubeck, and Michael I Jordan. Is Q-learning provably efficient? In Advances in Neural Information Processing Systems, pages 4863–4873, 2018.
> > > ```

---

### Official Review · Reviewer_1yNk · 2022-07-10

**Rating:** 6
**Confidence:** 4
**Soundness:** 3 good
**Presentation:** 2 fair
**Contribution:** 3 good

**Summary:**

This paper studies the problem of learning Nash equilibria in congestion games and in independent Markov congestion games, where the latter are a simple and natural generalization of congestion games to the Markov setting. The authors propose both centralized and decentralized algorithms for congestion games, whose sample complexity does not depend on the number of players’ actions (which in this setting can be exponential in the size of the game representation). The algorithms work under both semi-bandit feedback (the players observe the cost of each facility they choose) and bandit feedback (the players only observe the overall cost of all the facilities they select). Then, the authors generalize their centralized algorithms to the independent Markov congestion game setting, obtaining sample complexities that do not depend on the number of players’ actions.


**Questions:**

1) Can you provide me some intuition as to why you techniques cannot be used if one wants to minimize the regret of each player rather than the (overall) Nash regret as you are doing?


**Limitations:**

I think there are not limitations and potential negative societal impact for this work.

**Strengths And Weaknesses:**

ORIGINALITY

(+) The paper provides the first algorithms for learning Nash equilibria with bandit feedback in congestion games whose sample complexities guarantees do not depend on the number of players’ actions.

(-) The tools used in order to derive the main results in the paper are rather straightforward adaptations of commonly-used techniques: the centralized algorithms work by exploiting the optimism in face of uncertainty principle, while the decentralized ones leverage the Frank-Wolfe method, the G-optimal design, and the specific structure of rewards in congestion games.

QUALITY

(+) As far as I am concerned, the technical results are correct and adequately commented.

CLARITY

(+) The quality of writing is sufficiently good, even though there are some typos and grammatical errors that I invite the authors to spot by a simple applications of free spell checking tools.

(-) The presentation of the results is hard to follow in some parts. I think the main problem is that more than half of the paper is devoted to introducing models and notation, while only the last three pages are reserved for the main technical results. I think that maybe the authors should try to amend this issue by removing some notation that is not needed in the main paper and putting it in the appendix.

SIGNIFICANCE

(+) The results presented in the paper are significant since they considerably improve over already known sample complexities results for learning Nash equilibria with bandit feedback in congestion games.

(-) One of the main drawback (in my opinion) of the paper is that the decentralized algorithms do not extend to independent Markov congestion games. I think this is a major missing point and I am questioning whether it is worth devoting so much space in the paper to them even if the main decentralized algorithms do not work in that setting. Maybe it is better to concentrate on the simple congestion games model where all the algorithms are shown to work.

(-) The results can be of interest only for a small portion of the NeurIPS community, in particular to those interested in algorithmic game theory and online learning.

---

> ### Author Response · Authors · 2022-08-01
> **Response**
>
> Thank you very much for your careful review and constructive suggestions! Please find our response to your questions and concerns below:
> - **Lack of novelty in techniques:** We agree that G-optimal design and Frank-Wolfe are not new. However, in our decentralized algorithm, we propose a novel combination of Frank-Wolfe and G-optimal design that gracefully utilizes the structure of a congestion game. This idea may have uses in other problems.
> - **Lengthy notation part:** We will try to move some of the notation (especially those for Markov games) to the appendix in the final version.
> - **Extension for independent Markov congestion games (IMCGs):** We agree with the reviewer that the primary focus of this paper should be on the simple congestion game, which is also what we intend. We will reorganize the paper in the final version so that more space in the main content is devoted to the simple congestion game. Here, the centralized algorithm for IMCGs serves as a starting point for extending congestion games to Markov games to allow modeling non-stationarity, and we believe extending the decentralized algorithm is a promising future direction.
> - **Community interest:** Since our paper focuses on learning in games, people in the large multi-agent RL community may find it interesting. Meanwhile, we also believe the community of algorithmic game theory and especially online learning in NeurIPS is not small.
> - **Challenge of minimizing each player's regret:** The existing online Frank-Wolfe analysis depends on the $\ell_2$-Lipschitz constant of the function [Hazan, Kale 2012], which would lead to polynomial dependence on $A$. We didn't use this result because we want our complexity only depends on $m$ and $F$ instead of $A$, which can be exponentially large. One interesting future direction would be to design a no-regret algorithm that can converge to Nash Equilibrium with only polynomial dependence on $m$ and $F$.
>
> ```
> [Hazan, Kale. (2012)] Elad Hazan, and Satyen Kale. "Projection-free online learning." arXiv preprint arXiv:1206.4657 (2012).
> ```

---

> > ### Comment · Reviewer_1yNk · 2022-08-08
> > **Thanks for the answers**
> >
> > Thank you very much for your detailed response! You answered my curiosities. I will leave my score unchanged. If the paper is accepted, I also encourage the authors to make the modifications that they are mentioning in the response in the final version of the paper.

---

### Official Review · Reviewer_P7XP · 2022-07-17

**Rating:** 7
**Confidence:** 3
**Soundness:** 3 good
**Presentation:** 3 good
**Contribution:** 3 good

**Summary:**

This paper studies congestion games which has wide application in practice. The authors propose novel algorithms for learning the congestion game with semi-bandit or bandit feedbacks, i.e., the centralized algorithm and the decentralized algorithm. They define a new problem, Markov congestion game, and propose a centralized algorithm to learn the Markov congestion game. They further derive the upper bounds of the learning regret and the sample complexity for the above problems, which are independent of the size of the action set.

**Questions:**

Please see the above section.

**Limitations:**

Yes

**Strengths And Weaknesses:**

This paper studies the congestion game learning with (semi-)bandit problems, whose regret and sample complexity are not well understood before. Novel algorithms different from the existing UCB-type method are proposed to solve the corresponding problem. And the theoretical results are novel, which does not depend on the size of the action set and thus avoids the curse of exponential action set. Therefore, the theoretical contribution of this submission is significant. Moreover, this paper is well written and the structure is clear to the readers. Besides these strengths, I also have several questions regarding this paper:

(1) Can we still propose a UCB-type algorithm for the decentralized case? If not, what is the major obstacle of designing the UCB-type algorithm and what is the technical advantage of using FW + G-optimal design to solve this problem comparing to the UCB-type algorithm?

(2) The sample complexity for the algorithms with FW still has high-order dependence on m and F. Is it possible to lower the order of powers in the dependence on m and F?

---

> ### Author Response · Authors · 2022-08-01
> **Response**
>
> Thank you very much for your careful review and recognition of our contribution! Please find our response to your questions and concerns below:
> - **UCB-type decentralized algorithm:** Here, we cannot develop a UCB-type decentralized algorithm because from each player's perspective, the environment is non-stationary. That is, the marginal reward function for player $i$, $r_i^k(a_i):=\mathbb{E}\_{a_{-i}\sim\pi^k_{-i}}[r_i(a_i, a_{-i})]$, is changing from episode to episode. Therefore, it will be difficult for each player alone to maintain a confidence interval for her reward function and act according to it.
> - **Sample complexity of Frank-Wolfe:** The high-order dependence on $m$ and $F$ comes from the high-order dependence on $T$ in regret, which is a consequence of the repeated sampling step at line 4-5 in Algorithm 2. This step is necessary for our current analysis since we need to control the variance of the gradient estimator. However, we believe reducing this high-order dependence is a promising future direction.

---

### Comment · Area_Chair_YxgU · 2022-08-05
**Some questions to the authors**

I would like to thank both authors and reviewers for engaging in a constructive discussion!

From my side, after my own reading of the paper and the discussion so far, I have some remarks and questions, which I wanted to raise before the reviewer-metareviewer phase in order to include the authors in every part of the discussion.

1. The reviewers raised several concerns regarding the clarity of the authors' presentation. I agree that the notation and structure of the paper is, at times, hard to follow (for example, it is difficult to understand what's going on when Algorithms 3 and 4 are not even embedded in the main paper). I was expecting a revised version during the rebuttal phase (so the committee could assess the changes made), but this didn't take place: could the authors provide a more detailed account on the changes they intend to undertake?

1. What the authors call the "Nash regret" seems to be very closely related to the Nikaido-Isoda (NI) function [Nikaido H, Isoda K. "_Note on noncooperative convex games_", Pacific Journal of Mathematics, 1955, 5: 807–815; see also Raghunathan et al., "_Game Theoretic Optimization via Gradient-based Nikaido-Isoda Function_", ICML 2019], essentially up to replacing a max with a sum, and aggregating over episodes. Is my reading correct here?

1. The type of convergence guarantees provided by the authors' main results (Theorems 1-3) are not very clear to me. Specifically, if my reading is correct, it would seem that the only guarantee for the policy chosen by the players at each episode is that, out of $K$ episodes, some $\pi^k$ will be $\mathcal{O}(K^{-1/6})$-close to Nash equilibrium (in terms of the NI function). However, this is considerably weaker than claiming "convergence to Nash equilibrium" as the introduction would suggest - compare for example with the type of convergence results in the cited papers of Krichene et al. "_Online learning of Nash equilibria in congestion games_" and Héliou et al. "_Learning with bandit feedback in potential games_" which show that the _actual sequence of play_ converges (not some subsequence or a time-average). Especially given the close proximity of the title with the latter paper, I would have expected a much more in-depth analysis and comparison of results - am I missing something?

1. Finally, it is not clear why FW updates would be beneficial for reducing the dimensionality dependence in the semi-bandit case. The FW algorithm still needs to maintain an $\mathcal{A}$-size vector, so the algorithm is still subject to the curse of dimensionality. By contrast, an exponentially-weighted forecaster in the spirit of Cesa-Bianchi and Lugosi ("Combinatorial Bandits", 2012) would have the same dependence in terms of the $L^1$/$L^\infty$-norm, but without the $\mathcal{O}(A)$ per-iteration complexity, so it would seem to be a much better fit for the "update" template.

Thanks in advance for your input.

---

> ### Author Response · Authors · 2022-08-07
> **Response to Area Chair YxgU**
>
> Thank you very much for your very careful reading to our paper and constructive suggestions! Please find our response to your questions below:
> - **Paper revisions:** We made the following revisions to our paper and colored the added text to **red** in the paper:
>     - We added clarification on the notion of non-asymptotic convergence and sample complexity in Introduction section.
>     - We moved some motivating examples to the appendix.
>     - We added a remark about Nikaido-Isoda (NI) function below the definition of Nash regret.
>     - We moved the original Algorithm 3 (now Algorithm 2) into Section 4.1 of the main paper.
>     - We moved the notation for the Markov game to the section on independent Markov congestion games.
> Algorithm 4 is still in Appendix since we are not allowed to add one additional page now. We will move Algorithm 4 into the main paper in the final version where we can add the additional page.
> - **Nikaido-Isoda (NI) function:** Yes you are correct! The Nash regret is very closely related to the NI function in the sense that if we replace $\max_{i\in[m]}$ by $\sum_{i=1}^{m}$, the single-step Nash regret at episode $k$ becomes NI function evaluated at $\pi^k$. We adopt the notion of Nash regret as our objective from previous multi-agent RL papers such as [Ding et al. (2022)] and [Liu et al. (2021)] so that our results can be conveniently compared with theirs. We can conveniently switch to NI-type of Nash regret by multiplying our regret bounds by a factor of $m$, while our main conclusions will not be affected.
> - **Type of convergence guarantee:** Thanks a lot for pointing this out! We agree that asymptotically, our results imply a type of convergence weaker than those in [Krichene et al. (2015)] and [Heliou et al. (2017)].
>   - However, we want to emphasize that the focus in our paper is the finite-time analysis and the accumulative reward throughout the learning process, which has not been studied in the previous works to the best of our knowledge. We call an algorithm convergent if it has sublinear (Nash-)regret and this different notion of convergence is commonly used in the literature of online learning and reinforcement learning such as [Orabona. (2019)], [Ding et al. (2022)] and [Liu et al. (2021)]. We apologize for not making this point clear enough in previous introduction and we clarified in the revised paper. Meanwhile, we also want to point out that our results cannot be inferred from the types of convergence in [Krichene et al. (2015)] and [Heliou et al. (2017)] since they provide no finite-time guarantee.
>   - In addition to this, both [Krichene et al. (2015)] and [Heliou et al. (2017)] consider different settings compared with our work. In particular, [Krichene et al. (2015)] considered nonatomic congestion game (which is simpler than atomic congestion game we considered, as their potential function is convex) and proved  convergence in the sense of Cesaro means instead of actual sequence for discounted regret. They improved it to actual sequence convergence with an additional assumption that the discounting factor is changing appropriately. [Heliou et al. (2017)] proved actual sequence convergence but without convergence rate as well (they gave convergence rate under the very restrictive assumption that the sequence is converging to an almost deterministic Nash equilibrium). We believe our algorithms with action-dimension-free non-asymptotic convergence are important complements to their results.
> - **Action dimension dependence:** Sorry for the confusion. We claimed that our algorithms have no dependence on the number of actions in the sense of sample complexity (Nash regret) instead of computational complexity. Indeed we believe the exponentially-weighted forecaster in Cesa-Bianchi and Lugosi ("Combinatorial Bandits", 2012) also has $O(A)$ computational complexity per iteration as we need to sample from $A$ arms with corresponding probabilities. In addition, it is unclear if decentralized exponentially-weighted forecaster can have sublinear Nash regret that is independent of $A$. Reducing the computational complexity and making it independent of $A$ would be an interesting future direction.

---

> > ### Author Response · Authors · 2022-08-07
> > **References for the Response**
> >
> > ```
> > [Krichene et al. (2015)] Walid Krichene, Benjamin Drighès, and Alexandre M Bayen. Online learning of nash equilibria in congestion games. SIAM Journal on Control and Optimization, 53(2):1056–1081, 2015.
> > [Heliou et al. (2017)] Amélie Heliou, Johanne Cohen, and Panayotis Mertikopoulos. Learning with bandit feedback in potential games. Advances in Neural Information Processing Systems, 30, 2017.
> > [Orabona. (2019)] Francesco Orabona. A modern introduction to online learning. arXiv preprint arXiv:1912.13213, 2019.
> > [Ding et al. (2022)] Dongsheng Ding, Chen-Yu Wei, Kaiqing Zhang, and Mihailo R. Jovanovi ́c. Independent policy gradient for large-scale markov potential games: Sharper rates, function approximation, and game-agnostic convergence, 2022.
> > [Liu et al. (2021)] Qinghua Liu, Tiancheng Yu, Yu Bai, and Chi Jin. A sharp analysis of model-based reinforcement learning with self-play. In International Conference on Machine Learning, pages 7001–7010. PMLR, 2021.
> > ```

---

### Meta-Review · Area_Chair_YxgU · 2022-08-25

**Recommendation:** Accept
**Confidence:** Less certain

**Metareview:**

This paper examines the behavior of certain payoff-based learning algorithms in repeated congestion games.

The initial reviews were positive, but a more detailed reading of the paper and the discussion phase revealed the following issues:

- In terms of technical results, the authors are not proving convergence to Nash equilibrium, but the minimization of "Nash regret" (a regret-like variant of the Nikaido-Isoda function) which, at best, means that among $K$ episodes, there exists one episode for which $\pi^k$ is close to Nash. This issue was discussed during both the author-reviewer and reviewer-AC phases, and there was consensus that the authors' results cannot be labeled as "convergence to a Nash equilibrium". The authors' revision was not satisfactory in this regard: the statement that the paper is using a "non-asymptotic notion of convergence" misses the crux of the issue and creates more confusion so it was not seen as a step in the right direction.

- In terms of writing, the paper begins by treating repeated games in normal form, with polcies playing the role of mixed strategies; here, the episodes are single-shot instances of play. In the decentralized case however, the episodes are no longer single-shot instances of play, and the players are assumed to be using the same policy throughout this sampling period - in essence, playing the game in a stationary way. Finally, in Section 6, the authors treat stochastic / Markov games, where the notion of an episode is something still different. The inclusion of the algorithms in the revised makes things easier to follow, but it does not address why the UCB-based approach of the centralized case cannot be expected to work here - the authors are justifying the use of the FW update as a means to combat the curse of dimensionality but this does not explain why FW updates are not used in the relevant centralized updates as well. All this makes the paper quite difficult to follow at a technical level.

- Finally, in terms of positioning, the authors seem to be motivated by the recent regret-based works of Ding et al. (2022) and Liu et al. (2021) but, at the same time, they seem to ignore the much wider literature on game-theoretic learning. This was also flagged as a cause for concern during the discussion phase.

Despite the above shortcomings, the reviewers appreciated the paper's technical contributions and felt that they warrant acceptance. As a result, a decision was reached to make a "conditional accept" recommendation subject to the authors' revising their paper to account for the above issues. Specifically:
- The final version of the paper must make clear that the type of results obtained do not concern convergence to (or "learning of") Nash equilibrium, but the minimization of a regret-like measure based on the Nikaido-Isoda function. [In particular, the motivating question in L56 should be removed, and the abstract and introduction must be likewise amended]
- The passage from single-stage to multiple-stage episodes must be justified, as well as the passage from repeated to stochastic congestion games.
- The authors must improve the positioning of their paper in relation to existing works on multi-agent learning with bandit feedback; some relevant references are provided below.

These changes are quite extensive, but the quality of the paper did improve during the revision phase, so the committee felt confident that the authors can undertake the above required changes for the camera-ready submission.

**Relevant references:**
1. Sebastian Bervoets, Mario Bravo, and Mathieu Faure, Learning with minimal information in continuous games, Theoretical Economics 15 (2020), 1471–1508.
1. Mario Bravo, David S. Leslie, and Panayotis Mertikopoulos, Bandit learning in concave N-person games, NeurIPS ’18: Proceedings of the 32nd International Conference of Neural Information Processing Systems, 2018.
1. Roberto Cominetti, Emerson Melo, and Sylvain Sorin, A payoff-based learning procedure and its application to traffic games, Games and Economic Behavior 70 (2010), no. 1, 71–83.
1. Pierre Coucheney, Bruno Gaujal, and Panayotis Mertikopoulos, Penalty-regulated dynamics and robust learning procedures in games, Mathematics of Operations Research 40 (2015), no. 3, 611– 633.
1. Angeliki Giannou, Emmanouil Vasileios Vlatakis-Gkaragkounis, and Panayotis Mertikopoulos,
The convergence rate of regularized learning in games: From bandits and uncertainty to opti- mism and beyond, NeurIPS ’21: Proceedings of the 35th International Conference on Neural Information Processing Systems, 2021.
1. David S. Leslie, Reinforcement learning in games, Ph.D. thesis, University of Bristol, 2004.
1. David S. Leslie and E. J. Collins, Individual Q-learning in normal form games, SIAM Journal on Control and Optimization 44 (2005), no. 2, 495–514.
1. David S. Leslie and E. J. Collins, Generalised weakened fictitious play, Games and Economic Behavior 56 (2006), no. 2, 285–298.

**Award:**

No

---

### Decision · Program_Chairs · 2022-09-14

Accept